# Genetics of monozygotic twins reveals the impact of environmental sensitivity on psychiatric and neurodevelopmental phenotypes

Individual sensitivity to environmental exposures may be genetically influenced. This genotype-by-environment interplay implies differences in phenotypic variance across genotypes, but these variants have proven challenging to detect. Genome-wide association studies of monozygotic twin differences are conducted through family-based variance analyses, which are more robust to the systemic biases that impact population-based methods. We combined data from 21,792 monozygotic twins (10,896 pairs) from 11 studies to conduct one of the largest genome-wide association study meta-analyses of monozygotic phenotypic differences, in children, adolescents and adults separately, for seven psychiatric and neurodevelopmental phenotypes: attention deficit hyperactivity disorder symptoms, autistic traits, anxiety and depression symptoms, psychotic-like experiences, neuroticism and wellbeing. The proportions of phenotypic variance explained by single-nucleotide polymorphisms in these phenotypes were estimated ($h^2 = 0–18\%$), but were imprecise. We identified 13 genome-wide significant associations (single-nucleotide polymorphisms, genes and gene sets), including genes related to stress reactivity for depression, growth factor-related genes for autistic traits and catecholamine uptake-related genes for psychotic-like experiences. This is the largest genetic study of monozygotic twins to date by an order of magnitude, evidencing an alternative method to study the genetic architecture of environmental sensitivity. The statistical power was limited for some analyses, calling for better-powered future studies.

Complex phenotypes are likely to be affected by genetic and environmental factors and their interactions. Interactions between genetic variants and the environment increase phenotypic variability[1,2], which may be reflected in differences in the mean and/or variance of a phenotype in a population. This is evidenced when a genotype is associated with phenotype levels only under certain environmental conditions. Environmental sensitivity can also increase the variance of a trait if a genotype produces a wide range of phenotypes depending on environmental exposures, which may or may not also affect its population mean. Genetic knowledge of environmental sensitivity is most consistently exploited in bioengineering and evidenced in behavioural ecology, but it has been extremely challenging to evaluate in humans, especially for psychiatric disorders[3]. Understanding the genetic basis of environmental sensitivity is crucial for improving human health, as it informs on the biological pathways implicated in variations in responses to environmental exposures.

✉ e-mail: elham.1.assary@kcl.ac.uk

**Fig. 1 | GWAS of MZ differences approach.** These analyses are conducted in three main steps: First, a quantitative phenotype value is obtained for a population of MZ twin pairs. Second, the absolute phenotypic difference score is calculated for each MZ pair. This score reflects phenotypic differences due to environmental effects, as the environment makes genetically identical twins diverge phenotypically. The absolute phenotypic difference is corrected for age, sex, ten genetic principal components and any study-specific covariates. The residuals are standardized and inverse rank transformed. Third, a GWAS of the MZ differences score is conducted, using the phenotypic differences score for one twin from each pair and their genotype data; therefore, the sample comprises unrelated individuals.

In contrast with most GWASs, which estimate associations of genetic variants and phenotypic means, Genome-wide variance quantitative trait locus (vQTL) analysis[4,5] aims to discover genetic variants associated with phenotypic variance, which can be prioritized for a statistical test of gene–environment interaction[6]. However, phenotypic variance may be affected not only by gene–environment interactions[1], but also by selection[7], phantom vQTLs[8] and epistasis[4]. It is therefore challenging to robustly determine which potential mechanisms have given rise to the observed trait variance associated with a vQTL. Furthermore, commonly used population-based methods for estimating genetic effects using unrelated individuals suffer from variance inflation[9], bias due to insufficient correction of demographic and indirect genetic effects[10], and unstable test statistics for vQTLs when tested loci are in linkage disequilibrium with additive effects[11]. Although statistical correction for certain observed demographic effects (for example, age and sex) is possible, unobserved confounders (for example, residual population stratification, dynastic effects via parents, and assortative mating) cannot be corrected for.

GWASs of monozygotic (MZ) twin differences provide an alternative, family-based approach to estimating vQTLs that is less susceptible to these sources of bias[12] and can therefore more reliably identify variants that reflect environmental sensitivity.

MZ twins are nearly identical genetically. Therefore, within-pair phenotypic differences are probably due to chance or the environment[13]. Somatic mutations may also play a role in MZ differences for chromosomal and rare diseases[14], but are unlikely to play a systematic role in common and polygenic complex traits[15]. Although all MZ twin pairs have the same degree of genetic similarity, they have varying degrees of phenotypic similarity. Greater within-pair differences in a population of MZ twins therefore reflect the pairs' greater sensitivity to their non-shared environments. Jinks and Fulker[16] proposed a test in MZ twins for which the association between MZ pair differences and MZ pair mean score is obtained, and they provided a full description of its standard biometrical terms. The rationale behind their test of genotype–environment interaction is the same one that underlies tests involving inbred lines of animals that detect genotype–environment interaction through the heterogeneity of within-strain variances. We may now take single-nucleotide polymorphisms (SNPs) as measured genotype indicators and test for associations with within-pair differences.

A GWAS of MZ phenotypic differences can identify the loci associated with variations in environmental sensitivity while also controlling for dynastic and epistatic effects, which are difficult to account for in population-based approaches (Fig. 1 provides a schematic of a GWAS of MZ differences). Although this approach has been advocated for because it facilitates understanding of the genetics of environmental sensitivity[12], the requirement for a large sample of MZ twins has been a major impediment to progress in this field. In this Article, we report the findings from a GWAS meta-analysis of MZ differences for seven psychological phenotypes, using data from up to 21,792 MZ twins

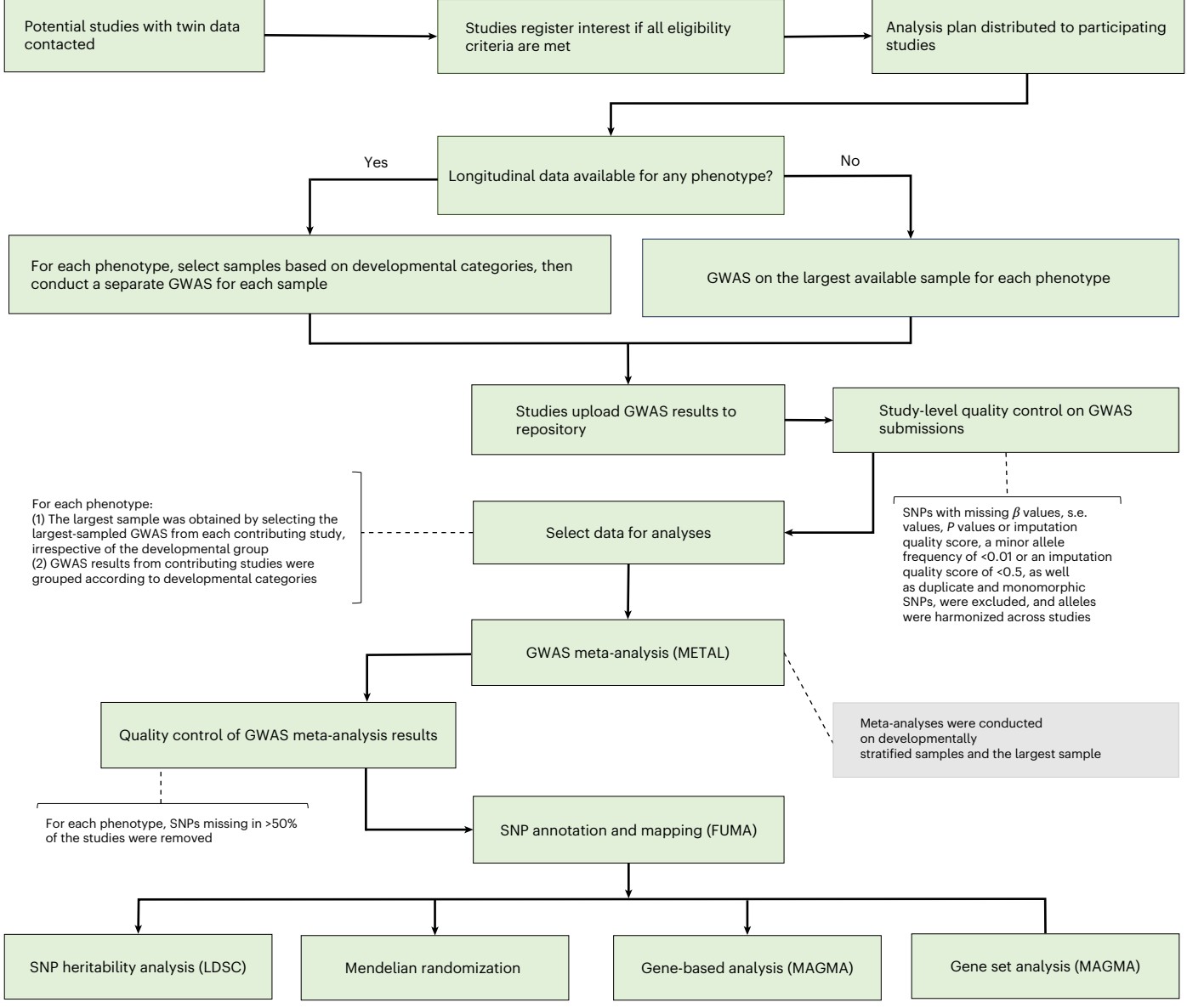

**Fig. 2 | Flow chart of the current study.** Flow chart genome-wide association meta-analysis of MZ twin differences in the present study. The eligibility criteria and developmental categories are described in the Methods. If multiple data points were available during the same developmental stage (for example, ages 7, 8 and 11 years), the largest sample was selected. Total *n* values per phenotype (representing the total across participating studies) were as follows: 13,738 (ADHD largest), 13,738 (ADHD child), 7,840 (ADHD adolescent), 12,354 (anxiety largest), 10,494 (anxiety child), 7,932 (anxiety adult), 14,152 (autistic traits largest), 13,130 (autistic traits child), 6,050 (autistic traits adult), 21,792 (depression largest), 10,510 (depression child), 18,074 (depression adult), 8,900 (neuroticism largest), 3,636 (psychotic-like experiences largest) and 13,740 (wellbeing largest). *n* values for GWAS analyses were halved as only data from one twin from each pair were included.

(10,896 pairs) from 11 studies. This is the largest GWAS conducted on MZ twin differences to date, representing an order of magnitude more participants than two previous MZ twin differences GWAS[17,18]. We conducted meta-analyses separately for children, adolescents and adults and identified 13 genome-wide significant associations across the phenotypes studied. This enabled us to estimate the SNP heritability (that is, the proportion of a trait's variance that can be attributed to genetic variation explained by SNPs) of environmental sensitivity to various mental health phenotypes (adolescent attention deficit hyperactivity disorder (ADHD): $h^2 = 0.18$ and s.e. = 0.11; child ADHD: $h^2 = 0.04$ and s.e. = 0.06; adult autistic traits: $h^2 = 0.09$ and s.e. = 0.15; depression: $h^2 = 0.03$ and s.e. = 0.09). We also found that higher genetic liability to depression was associated with greater environmental sensitivity to depression in adults.

## Results

Our empirical analyses included a GWAS meta-analysis, gene-based and gene set analyses and Mendelian randomization analyses (Fig. 2 provides a flow chart for the study). Authors of contributing studies were asked to conduct GWASs of MZ differences separately for samples from child, adolescent and adult twins if repeated measures across lifespan were available. The study-level GWAS results were subjected to quality control and harmonization (Supplementary Information section 2.2) using EasyQC (version 23.8)[19], then meta-analysed using the inverse-variance-weighted fixed-effect meta-analysis method in METAL (2011 release)[20]. GWAS meta-analyses were conducted using the largest available sample from each study and separately in developmentally stratified samples (Methods). GWAS meta-analysis results were quality controlled (Methods), then used for gene-based

**Table 1 | Descriptive statistics of the samples in the current study**

| Phenotype | Sample | $n_{studies}$[a] | $n_{MZ\ twins}$[b] | $n_{female}$ | Mean age (years) | Mean $r_{MZ}$[c] |
|---|---|---|---|---|---|---|
| ADHD symptoms | Largest[d] | 4 | 13,738 | 7,420 | 9.54 | 0.71 |
| | Child[e] | 4 | 13,738 | 7,420 | 9.54 | 0.71 |
| | Adolescent[f] | 4 | 7,840 | 4,821 | 20.28 | 0.58 |
| Anxiety symptoms | Largest | 5 | 12,354 | 6,802 | 20.40 | 0.52 |
| | Child | 3 | 10,494 | 5,881 | 10.01 | 0.54 |
| | Adult[g] | 5 | 7,932 | 5,187 | 33.35 | 0.46 |
| Autistic traits | Largest | 4 | 14,152 | 7,779 | 15.79 | 0.63 |
| | Child | 3 | 13,130 | 7,084 | 10.22 | 0.69 |
| | Adult | 3 | 6,050 | 3,567 | 22.53 | 0.63 |
| Depression symptoms | Largest | 11 | 21,792 | 13,011 | 43.23 | 0.41 |
| | Child | 3 | 10,510 | 5,726 | 10.05 | 0.51 |
| | Adult | 11 | 18,074 | 11,780 | 43.62 | 0.38 |
| Neuroticism | Largest | 4 | 8,900 | 4,096 | 35.05 | 0.35 |
| PLEs | Largest | 2 | 3,636 | 2,135 | 15.91 | 0.46 |
| Wellbeing | Largest | 9 | 13,740 | 8,009 | 41.30 | 0.31 |

[a]$n_{studies}$ represents the number of contributing datasets for each phenotype. [b]GWAS $n$ was halved as only genotype data from one twin from each pair were used.[c]Mean $r_{MZ}$ represents the mean MZ twin correlation across studies. [d]For ADHD symptoms, the largest sample was the sample from children. In general, this category represents the largest available sample, which was obtained by selecting the largest sample from each study, irrespective of age group. [e]The child category includes data from studies in which participants were aged 5–12 years.[f]The adolescent category includes data from studies in which participants were aged 13–18 years. [g]The adult category includes data from studies in which participants were aged >18 years. Data from individual studies are reported in the Supplementary Data.

and gene set analyses using MAGMA (version 1.08) in the FUMA web application (version 1.5.2)[21] and SNP heritability analyses using LDSC (version 1.0.1)[22]. We used Mendelian randomization to estimate the effects of psychological phenotypes (as reflected in GWAS associations with means) on phenotypic variance (Methods).

### Descriptives
Table 1 shows descriptive statistics of the GWAS meta-analysis samples per phenotype. Supplementary Information Section 1.1 and the Supplementary Data present descriptives for all of the participating studies. The largest sample was for depression symptoms, with 21,792 MZ twins from 11 studies. The smallest was for psychotic-like experiences (PLEs; 3,636 twins from two cohorts). Mean MZ correlations across studies ranged between $r = 0.31$ for wellbeing and $r = 0.71$ for child ADHD. Overall, within-twin MZ correlations tended to be lower in samples from adult twins than those from child twins.

### MZ GWAS meta-analysis
Meta-analyses for each phenotype identified two genome-wide significant variants (Table 2): one associated with variability in wellbeing (rs2940988; $P = 9.93 \times 10^{-9}$), located in the intronic region of the protein-coding chromosome 4 open reading frame 19 (C4orf19) gene; and one (rs60358762; $P = 5.07 \times 10^{-9}$) associated with variance in anxiety symptoms in adults, located in the intergenic region of the protein-coding SLC15A1 gene on chromosome 13. Manhattan plots, quantile–quantile (QQ) plots and the genomic regions of these genome-wide significant variants are presented in Fig. 3. No variants were genome-wide significantly associated with variance in the other phenotypes (Supplementary Table 4 provides the top SNPs for each phenotype). Supplementary Figs. 7–11 provide Manhattan and QQ plots for all of the tested phenotypes.

### Gene-based and gene set analyses
The MAGMA gene-based analysis (Methods) found several genes associated with phenotypic variability; however, only two associations passed the threshold for Bonferroni correction for multiple testing (Table 2). The patched 1 (PTCH1) gene was associated with variance in depression ($P = 1.80 \times 10^{-6}$) and the chromosome 15 open reading frame

38 (C15orf38) gene was associated with variance in anxiety symptoms ($P = 2.00 \times 10^{-7}$). Supplementary Figs. 12–16 and Supplementary Table 6 provide the top genes per phenotype.

The MAGMA competitive test of gene sets (Methods) identified nine significant associations after Bonferroni correction for multiple testing (Table 2). Two gene sets were significantly associated with variance in depression symptoms, two with neuroticism, three with PLEs and two with autistic traits (one in samples from adults and one in samples from children). Supplementary Table 7 provides the top gene sets per phenotype, Supplementary Tables 8–11 provide details of significant gene sets and Supplementary Table 12 provides biological annotations for all genome-wide significant results.

### SNP heritability
We estimated the SNP heritability of MZ differences using LDSC (version 1.0.1)[22] (Methods). The SNP heritability estimate of environmental sensitivity to ADHD symptoms in the samples from adolescents was 0.18 (s.e. = 0.11); the estimate in the samples from children was 0.04 (s.e. = 0.06). The SNP heritability estimate for environmental sensitivity to adult autistic traits was 0.09 (s.e. = 0.15) and for depression symptoms it was 0.03 (s.e. = 0.09 in children and s.e. = 0.06 in adults). All estimates, including those from the remaining phenotypes, were not statistically significant (Supplementary Table 13). We could not estimate the genetic correlation ($r_g$) between all phenotypes because the SNP heritabilities were too low and imprecise, except for child and adolescent symptoms of ADHD ($r_g = 0.82$; s.e. = 0.56; $P = 0.15$). Overall, despite this being one of the largest samples of MZ twins to date, low power resulted in large confidence intervals around the heritability estimates.

### Mendelian randomization
It has previously been speculated that environmental sensitivity might relate to polygenic liability rather than single loci, due to the environment interacting with a polygenic biological component[23]. We used Mendelian randomization to estimate the influence of the genetic liability of psychological phenotypes on their environmental sensitivity (Methods). We found a strong effect for depression ($\beta = 0.84$; s.e. = 0.26; $P = 0.002$). We also ran the analyses separately for our samples from

**Table 2 | Genome-wide significant results**

**GWAS meta-analyses**

**SNP based analysis**

| Phenotype | Sample | SNP | Chromosome | Position | Gene | A1 | EAF | β | s.e. | P | n | Effect across studies[a] |
|---|---|---|---|---|---|---|---|---|---|---|---|---|
| Anxiety | Adult | rs60358762 | 13 | 99,411,217 | *SLC15A1* | A | 0.03 | 0.44 | 0.1 | $5.07\times10^{-9}$ | 3,033 | +++?? |
| Wellbeing | Adult | rs2940988 | 4 | 37,586,376 | *C4orf19* | T | 0.88 | 0.16 | 0.03 | $9.93\times10^{-9}$ | 6,464 | ++?++−+++ |

**Gene-based analysis**

| Phenotype | Sample | Gene | Chromosome | $n_{SNPs}$ | $n_{parameters}$ | n | Z statistic | P |
|---|---|---|---|---|---|---|---|---|
| Anxiety | Largest | *C15orf38* | 15 | 28 | 7 | 5,265 | 5.08 | $2.00\times10^{-7}$[b] |
| Depression | Largest | *PTCH1* | 9 | 130 | 12 | 10,166 | 4.63 | $1.80\times10^{-6}$[c] |

**Gene set analysis**

| Phenotype | Sample | Gene set | $n_{genes}$ | β | s.d. | s.e. | P | $P_{Bonferroni}$[d] |
|---|---|---|---|---|---|---|---|---|
| Autistic traits | Child | Genes downregulated in embryonic fibroblasts upon stimulation with TGFβ1 for 1h | 5 | 1.68 | 0.03 | 0.34 | $6.00\times10^{-7}$ | 0.01 |
| | Adult | Regulation of protein localization to cilium | 7 | 1.43 | 0.03 | 0.26 | $1.00\times10^{-8}$ | 0.0002 |
| Depression | Child | Proteasome regulatory particle | 19 | 0.79 | 0.03 | 0.17 | $1.41\times10^{-6}$ | 0.02 |
| | | Proteasome degradation | 50 | 0.52 | 0.03 | 0.11 | $2.48\times10^{-6}$ | 0.04 |
| Neuroticism | Adult | Gemini of coiled bodies | 9 | 1.01 | 0.02 | 0.20 | $2.00\times10^{-7}$ | 0.003 |
| | | Negative regulation of vasculature development | 92 | 0.38 | 0.03 | 0.08 | $5.97\times10^{-7}$ | 0.01 |
| PLEs | Adolescent | Regulation of dopamine uptake involved in synaptic transmission | 8 | 1.37 | 0.03 | 0.28 | $5.00\times10^{-7}$ | 0.01 |
| | | Catecholamine uptake involved in synaptic transmission | 11 | 1.23 | 0.03 | 0.26 | $9.55\times10^{-7}$ | 0.01 |
| | | Extrinsic component of endoplasmic reticulum membrane | 5 | 1.63 | 0.03 | 0.35 | $1.59\times10^{-6}$ | 0.02 |

[a]A question mark indicates that the SNP was missing in the study, − or + sign indicate whether the direction of effect for each study was the opposite of, or consistent with the direction of effect from meta-analysis. [b]The Bonferroni-corrected *P* value significance threshold was $P=0.05/18,535=2.698\times10^{-6}$. [c]The Bonferroni-corrected *P* value significance threshold was $P=0.05/18,624=2.685\times10^{-6}$. [d]$P_{Bonferroni}$ represents the Bonferroni-adjusted *P* value for multiple testing correction. $P<5\times10^{-8}$ was the significance threshold used to adjust for multiple comparisons when identifying genome-wide significant SNPs. A1, effect allele; β, beta estimate from linear regression; EAF, average effect allele frequency across studies.

children and those from adults (Fig. 4). The signal for depression was driven by analyses in adults ($\beta = 1.58$; s.e. = 0.29; $P = 5\times10^{-7}$), with the childhood association attenuated ($\beta = 0.35$; s.e. = 0.35; $P = 0.36$), and these estimates differed significantly (interaction $P = 0.008$). There was little evidence for heterogeneity of effect estimates across depression liability instruments. Other phenotypes did not exhibit an influence of liability on environmental sensitivity after correcting for multiple testing (Supplementary Table 14). We found little evidence that mean body mass indices and years of schooling affected phenotypic variance. Since vQTL effects could be biased by main effects, we conducted simulations to evaluate the sensitivity of the analytical approach to the Mendelian randomization results being driven by this bias (Supplementary Information section 7). We found that the MZ differences model can be liable to this problem under some phenotype normalization approaches, but that this was less likely to be driving the results in this study because we normalized by inverse rank transformation (Supplementary Fig. 17).

## Data simulation

We used simulations to investigate the properties of the method used to detect MZ twin-based variance loci and compared them with those of population-based vQTL methods. First, we found that the MZ twin-based method has the greatest power when the narrow-sense heritability is highest (Fig. 5a) and the residual variance is lowest, and therefore when variance effects explain the larger fraction of the difference between MZ pairs, as was suggested previously[12]. Second, we found that for moderate heritability the MZ differences approach has substantially greater statistical power than the population-based approach when sample sizes are equal (that is, 10,000 MZ twin pairs using the MZ twin-based method versus 20,000 unrelated individuals using the population-based vQTL method). However, in practice, the number of population-based samples generally available drastically outstrips the number of MZ twin samples available. When using a more realistic sample size (for example, 10,000 MZ twin pairs versus 500,000 population samples), the MZ differences approach only achieves similar power to the population approach when narrow-sense heritability values are very high (for example, >0.9). It has recently been shown that a variant tested for interaction can have inflated test statistics when in linkage disequilibrium with a strong additive effect[11], and we investigated whether that mechanism can also adversely impact MZ twin-based estimates. Our simulations show that this problem is substantially exacerbated through population-based vQTL methods compared with direct interaction tests, but the MZ twin-based approach is robust to this bias (Fig. 5b).

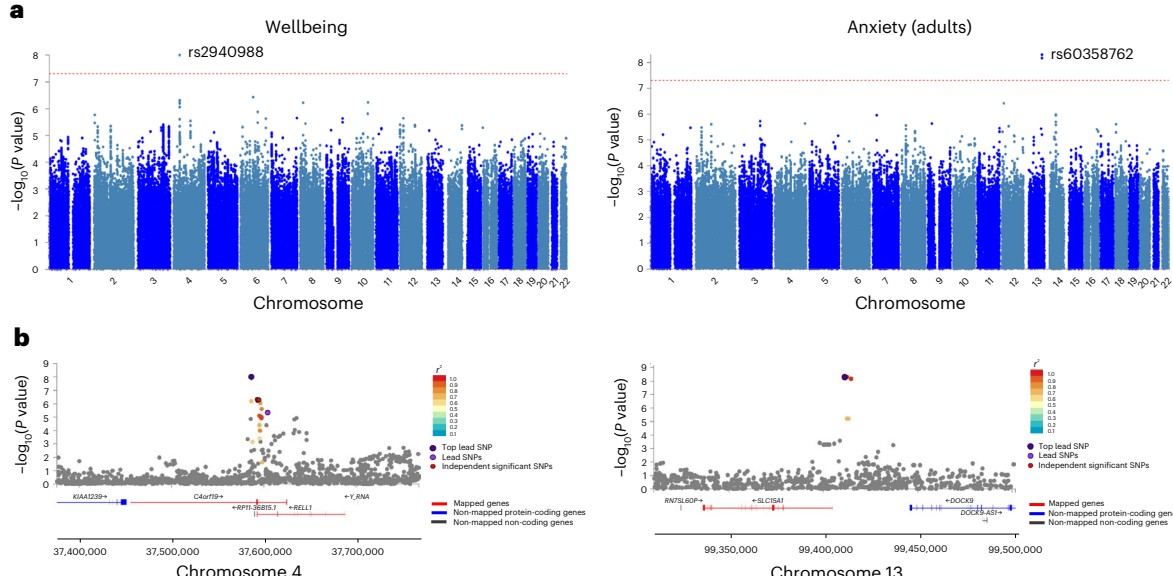

**Fig. 3 | Manhattan and regional plots of genome-wide significant SNPs.**
**a**, Manhattan plots based on GWASs of MZ differences for wellbeing (left) and anxiety in samples from adults (right). The red dashed lines indicate the genome-wide significant threshold of $P = 5 \times 10^{-8}$, corrected for multiple testing.

**b**, Regional plots of independent significant SNPs for wellbeing (left; rs2940988; $\beta = -0.16$; $P = 9.93 \times 10^{-9}$) and anxiety (right; rs60358762; $\beta = 0.44$; $P = 5.07 \times 10^{-9}$). SNPs not in linkage disequilibrium with significant independent lead SNPs in the selected region are coloured grey.

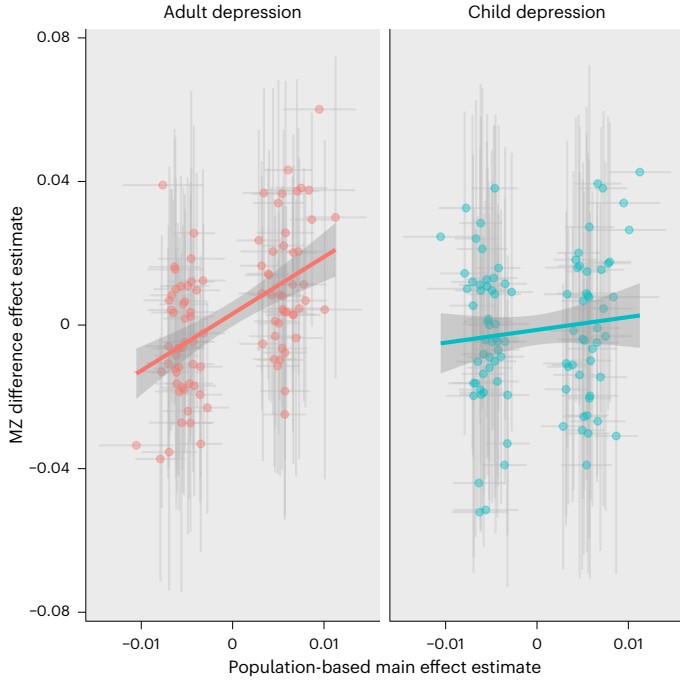

**Fig. 4 | Two-sample Mendelian randomization estimates of the effect of genetic liability to depression on variance in depression.** Genetic liability to depression, based on population studies is shown on the *x* axes and variance in depression scores (using 102 variants found to associate with depression incidence) is shown on the *y* axes. Genetic liability to depression increased the variability in depression outcomes and this was attenuated to the null in children (interaction term in linear regression $P = 0.008$). The error bars represent 95% confidence intervals of the variant effect estimates.

## Discussion

Several genome-wide significant results were notable, including our finding that the *PTCH1* gene was associated with variation in depression symptoms, as this gene has previously been reported to be associated with depression-related phenotypes, including neuroticism[24,25], anxiety[26], depression symptoms[24], feeling emotionally hurt[27] and sensitivity to environmental stress and adversity[27]. The *C15orf38* gene (also known as *ARPIN-AP3S2*) was associated with variance in anxiety symptoms in our samples from children and has previously been associated with type 2 diabetes in adults[25,28] and corticotropin-releasing factor protein levels[29], which are involved in regulating anxiety, mood, eating and inflammation[30]. Hypoglycaemia symptoms in type 2 diabetes include a rapid heartbeat, sweating and nervousness, all of which are physical sensations associated with anxiety. It is possible that certain variants in this gene impact sensitivity to the effects of diet and stressors that are involved in the variability in insulin[31], unpleasant physical sensations of which may be contextualized and made sense of as worries and anxieties (for example, the two-factor model of emotions[32]).

For autistic traits, the identified gene set included genes involved in tissue morphogenesis and healing that regulate the response to transforming growth factor beta (TGFβ1) levels and are involved in tissue repair pathways[33]. Growth factors serve important roles in neurodevelopment, immune function and development of the central nervous system and there is evidence that autism is associated with TGFβ1 and other growth factor-encoding genes[34–36]. For PLEs, the identified gene sets were related to the regulation of dopamine and catecholamine uptake. Our findings are supported by catecholamine's involvement in the stress response and the hypothesized role of dopamine system dysregulation in the aetiology of psychosis[37]. Since gene–environment interactions have been implicated in variations in PLEs[38], the association between the biological processes implicated herein and variability in psychotic experiences may partly be under the influence of the environment.

We also estimated the SNP heritability of environmental sensitivity (adolescent ADHD: $h^2 = 0.18$ and s.e. = 0.11; child ADHD: $h^2 = 0.04$ and s.e. = 0.06; adult autistic traits: $h^2 = 0.09$ and s.e. = 0.15; depression: $h^2 = 0.03$ and s.e. = 0.09), but the estimates were not statistically significant. We also showed that variants that affect mean levels of depression and anxiety influence the variance of these phenotypes. Several population-based methods for vQTL analysis are known to be susceptible to bias due to mean effects. In contrast, in principle, the MZ differences design is protected from this problem. Therefore,

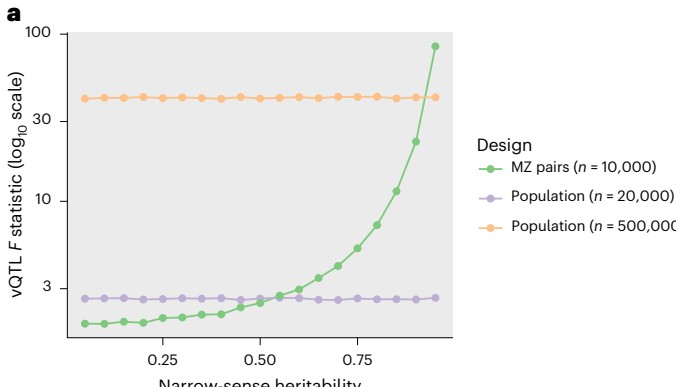

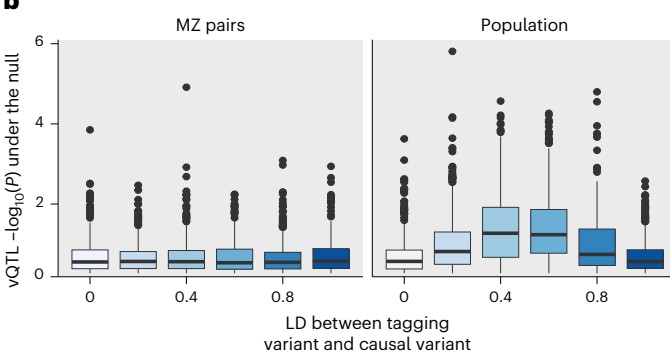

**Fig. 5 | The MZ differences approach complements population-based vQTL detection. a**, Comparison of the power between the MZ differences approach and the population-based approach (using the deviation regression model method) for detecting vQTLs. Comparisons were made between the MZ pairs approach and two different sample sizes for the population-based approach (one matching (with double the number of unrelated individuals; 20,000) and one with a typical sample size for modern GWASs (500,000)). The power of the MZ differences approach increases as the narrow-sense heritability increases. **b**, False discovery rates (*y* axis) due to incomplete linkage disequilibrium (LD) with a tagging causal variant (*x* axis), compared between the MZ differences approach (left) and population-based approach (right). The box plots represent interquartile ranges around median values of the simulations, with the whiskers representing ±1.5× the interquartile range and the points representing outliers. The presence of an additive causal variant tagging the tested SNP leads to elevated false discovery rates in the population approach, but not in the MZ differences approach.

our study provides independent evidence that mean effects can influence variance.

The main strength of our study is the use of the MZ differences method to investigate the genetics of environmental sensitivity in the largest sample of MZ twins and a wide range of psychological phenotypes from a developmental perspective. The main limitation is the limited statistical power for detecting small genetic effects on variance. The findings should therefore be considered in light of our statistical power. Furthermore, as with all empirical analyses, our inferences depend on specific assumptions that may not hold. First, we took the MZ differences score to reflect genuine phenotypic variability. MZ differences, however, also reflect measurement errors that are difficult to separate from genuine phenotypic differences, and it is unclear whether MZ differences are stable across time. Future projects should investigate this approach with other phenotypes with low measurement errors, such as height, and consider a longitudinal design to assess the stability of these differences. Second, it is challenging to determine which mechanisms explain phenotypic variance. Genetic sensitivity to the environment and epigenetic processes, such as DNA methylation, imprinting, chorionicity and skewed X inactivation

(for female MZ twins), also contribute to MZ phenotypic differences[15,39]. Also, the people included in our samples were all of European genetic ancestry since data from twins with DNA in other ancestries were not available in sufficient sample sizes. Our findings may not be generalizable to non-European genetic ancestries. The current study underlines the utility of DNA data from twins and should encourage funding for genetic data collection in multi-ancestry twin cohorts.

In summary, we have identified novel genetic factors associated with phenotypic variability. Our study illustrates the importance of large meta-analyses of genotyped MZ twin samples as a method for discovering and understanding the genetics of phenotypic variance and environmental sensitivity.

## Methods

This research complies with all of the ethical regulations related to the secondary analysis of data collected by various cohorts, each of which obtained ethical approval and informed consent.

### Ethical approval
**Danish Twin Registry.** The collection and use of biological material and survey information was approved by the Regional Committees on Health Research Ethics for Southern Denmark. This study is registered in Southern Denmrak University's internal list (notification number 11.059) and complies with the rules of the General Data Protection Regulation.

**Finnish Twin Registry.** Ethics approval was obtained for multiple studies, the latest of which was related to the transfer of all available DNA samples, genotypes and associated phenotypes to the THL Biobank by the Hospital District of Helsinki and Uusimaa ethics board in 2018 (1799/2017).

**Netherlands Twin Register.** Ethical approval was provided by the Central Committee on Research Involving Human Subjects of the VU University Medical Center in Amsterdam, an institutional review board certified by the US Office for Human Research Protections (IRB number IRB-2991 under Federal-wide Assurance-3703; IRB/institute codes, NTR 03-180).

**Murcia Twin Registry.** The procedures of this registry have been approved by the University of Murcia Research Ethics Committee.

**Older Australian Twins Study.** The Older Australian Twins Study (OATS) was approved by the human research ethics committees of the Australian Twin Registry, University of New South Wales, University of Melbourne, Queensland Institute of Medical Research (QIMR) and South Eastern Sydney and Illawarra Area Health Service. Written informed consent was provided by all participants.

**QIMR.** Studies were approved by the QIMR Berghofer Medical Research Institute Human Research Ethics Committee.

**Swedish Twin Registry.** The different twin studies received separate approvals from the regional ethical review board in Stockholm (Dnr 80:80, 84:61; 93:226, 98:319 and 2010/657-31/3 (Swedish Adoption/Twin Study of Aging); Dnr 98:380 (OCTO-Twin); and Dnr: 97:051 and 2007/151-31/4 (HARMONY). Ethical approval for the PSYCH, TwinGene and Young Adult Twin Study in Sweden cohorts was given by the Uppsala Ethical Review Authority (2019-06066).

**Twins Early Development Study (TEDS).** Ethical approval for TEDS has been provided by the King's College London Ethics Committee (reference: PNM/09/10–104). Written informed consent was obtained prior to each wave of data collection from parents and from twins themselves from age 16 onwards.

**TwinsUK.** Approval was obtained from the TwinsUK BioBank, approved by the North West–Liverpool Central Research Ethics Committee (reference 19/NW/0187; Integrated Research Application System ID = 258513). This approval supersedes earlier approvals granted to TwinsUK by the St Thomas' Hospital Research Ethics Committee and later by the London–Westminster Research Ethics Committee (reference EC04/015), which have now been subsumed within the TwinsUK BioBank.

## Study design

Figure 3 provides a flow chart for the current study. The eligibility criteria for inclusion of a study's data in the present study analyses included: (1) cohorts comprising at least 100 pairs of MZ twins; (2) genotyping data for one or both twins; (3) availability of imputed genotype data (for example, 1000 Genomes or Haplotype Reference Consortium (HRC) data); (4) complete data from both twins for one or more phenotypes (imputation of missing data for incomplete pairs is not recommended); (5) complete covariate data for both twins (that is, age, sex and principal components for the genotyped twin); and (6) samples of European ancestry. Many of the authors of the studies with contributing data are part of the Within Family Consortium.

The analysis plan and pipeline were written, pre-specified and shared with interested cohorts to enable them to conduct GWASs of MZ differences for their available phenotypes, then the results were uploaded to a designated repository (https://github.com/LaurenceHowe/MZTwins-vQTL).

For some participating studies, data were available across the lifespan of participants, thus comprising repeated measures for certain phenotypes. It was therefore possible to explore genetic associations in the context of development by conducting developmentally stratified analyses. The developmental groups were defined as childhood (5–12 years of age), adolescence (13–18 years of age) and adulthood (>18 years of age). GWAS analyses were conducted separately for different cohorts and for each developmental stage if data were available. For anxiety symptoms, depression symptoms and autistic traits, GWAS data were available for children and adults. For ADHD symptoms, GWAS data were available for children and adolescents. For wellbeing and neuroticism, the samples comprised adults only, and for PLEs the samples comprised adolescents only.

Two sets of meta-analyses were then conducted using the GWAS results: a developmentally informed analysis, whereby GWAS results across studies were grouped according to the developmental stage of the sample and meta-analyses were conducted per phenotype (for example, depression child or depression adult); and a developmentally agnostic analysis, whereby a meta-analysis was conducted for each phenotype using the largest sample from each study, regardless of the developmental stage (for example, depression largest or anxiety largest). This ensured maximum power for meta-analysis per phenotype via the largest $n$ value (Supplementary Information section 1.1 provides more details on study design).

## Samples

The samples analysed in this study included MZ twin pairs from cohort studies or twin registries in Australia and Europe (Supplementary Information Section 1.2 and Supplementary Table 1 provide details of the participating studies). Informed consent and ethical approvals were obtained for all participating cohorts (Supplementary Information). Table 1 shows the total sample sizes per phenotype and across developmental groups.

## Phenotypes

The MZ differences method requires continuous or categorical non-binary phenotypic data to calculate variance. Therefore, we used mean symptom scores instead of case/control diagnosis and the preference was for continuous measures. Various instruments have been developed for the assessment of psychological phenotypes, as was reflected in the participating studies. The scales differed in the numbers of items included, the types of symptoms assessed and the informant source. If multiple rating scales of a phenotype were available, study authors were asked to select the scale with the most items (tapping most symptom domains). Scales were coded so that higher values represented higher symptom levels. Absolute phenotypic differences were obtained for each MZ pair. Using linear regression, absolute phenotypic differences were corrected for age, sex, ten genetic principal components and any study-specific covariates. The residuals were standardized and rank transformed to be used as the phenotype in the GWASs. A brief description of each phenotype is provided below (Supplementary Information Section 1.1 gives details).

ADHD is a childhood-onset neurodevelopmental disorder of attention, activity and impulsivity. Symptoms commonly persist into adulthood. They are typically measured continuously using rating scales, often with separate scales for attention problems and hyperactivity or impulsivity, which can be summed to give a total score of ADHD symptoms.

Anxiety is heterogeneous, with clinical diagnoses comprising specific anxiety disorders (for example, phobias, post-traumatic stress disorder or social anxiety disorder) and generalized anxiety disorder. We were interested in generalized anxiety symptoms, usually measured via self-report, and reflected in a total score of anxiety symptoms.

Autism spectrum disorder is a neurodevelopmental disorder broadly reflecting difficulties in social interaction and verbal communication, as well as repetitive behaviours. Symptoms typically emerge in early childhood, and assessment is carried out via questionnaires and/or interviews. The continuous scores reflect the presence or extent of autism traits rather than a diagnosis of autism spectrum disorder.

Depression is heterogeneous, with many clinical presentations. A diagnosis requires a distinct change of mood, characterized by sadness or irritability and accompanied by psychophysiological changes, such as disturbances in sleep and appetite or loss of the ability to experience pleasure. The phenotypic scores for depression reflect the presence of any of these symptoms rather than a diagnosis of major depression.

Neuroticism is a personality domain and refers to a lack of emotional stability, stress vulnerability, the tendency to experience intense negative emotions, affects and cognitions, and impulsive behaviours under emotional strain. Neuroticism is considered a risk factor for anxiety and depression.

PLEs include a sub-clinical threshold of symptoms related to psychosis or schizophrenia disorders, such as persecutory ideation or perceptual abnormalities, that are prevalent in the community and non-clinical samples. PLEs are screened using self- or other-report questionnaires or interviews that cover some or all of these domains: paranoia, hallucinations, cognitive disorganization, anhedonia and negative symptoms.

Wellbeing includes both hedonic and eudaimonic wellbeing and is typically assessed via questionnaires that index an individual's subjective sense of wellness, such as reporting satisfaction with one's life or being hopeful and optimistic about it. The data from participating studies mainly related to subjective wellbeing (for example, life satisfaction). We preferred to use data for which wellbeing had been measured using a battery of questions. If data were only available from a single question reported on a Likert scale, the response variable was treated as a continuous scale.

## Genotypes

Participating studies were required to have genotype data from all 22 autosomes imputed to either the 1000 Genomes reference panel (preferably phase 3) or the HRC. Almost all contributing studies had already participated in a project related to the Within Family Consortium[10] and used the same protocol in the automated scripts for genetic data preparation and quality control procedures before GWAS analysis. Minimum

quality control requirements at the study level included: filtering SNPs for an imputation quality of >0.3 for HapMap imputed data or >0.5 for 1000 Genomes or HRC data; a call rate of >95%; and a minor allele frequency of >1%. Studies also removed one pair randomly when there were two MZ pairs with a kinship of >0.1. Study-level genotyping and quality control information are included in the Supplementary Data.

## Analyses

**Simulations.** We investigated the statistical properties of the MZ differences GWAS method using simulations. Methods using family-based design, such as the MZ differences method, complement population-based vQTL methods in three ways: (1) statistical power; (2) robustness to bias due to additive effects; and (3) providing an alternative identification strategy for triangulation[40]. To simulate vQTLs, we used the following data-generating model.

$$y_{i,t} = \alpha + \beta_1 G_i + z_i + \upsilon_{i,t} + e_{i,t}$$

where $y_{i,t}$ is the phenotypic value for twin $t = \{A,B\}$ at MZ pair $i = \{1,...,n\}$ and $\alpha$ is an intercept term, which is set to 0 for simplicity. $\beta_1$ is the additive effect of the SNP $G_i$, which is distributed as

$$G_i \sim \text{Binom}(2,p)$$

where $p$ is the allele frequency. Note that $G_i$ is the same for both twins in the $i$th MZ pair. $z_i$ is the remaining polygenic risk for the MZ pair, which is normally distributed with a mean of 0 for simplicity, and variance is defined as

$$z_i \sim n\left(0, \sigma^2_g - 2p(1-p)\beta_1^2\right)$$

where $\sigma^2_g$ is the genetic variance of the trait. The variance heterogeneity term $\upsilon_{i,t}$ is modelled as

$$\upsilon_{i,t} \sim n(0, \beta_2 G_i)$$

such that each additional allele at $G$ increases the within-MZ difference by a factor of $\beta_2$. Finally, $e_{i,t}$ is the independent individual residual variance defined as

$$e_{i,t} \sim n\left(0, 1 - \sigma^2_g - \sigma^2_v\right)$$

where $\sigma^2_v$ is the variance of $\upsilon_{i,t}$. Therefore, for a pair (A and B) of MZs, the additive genetic factors $G_i$ and $z_i$ are fixed, but the within-MZ variability is induced from the $\upsilon_{i,t} + e_{i,t}$ terms. We estimated vQTL effects from unrelated individuals (choosing one MZ at random) using the deviation regression model from Marderstein and colleagues[41]. We estimated vQTL effects using MZs and the following MZ difference model:

$$|y_{i,A} - y_{i,B}| = \hat{\beta}_2 G_i + \epsilon_i$$

where $\hat{\beta}_2$ is an estimate of the vQTL effect, and $\epsilon_i$ is the residual error from this regression. We investigated the power of each method by generating vQTL effects ($\beta_2$) calibrated to have 80% statistical power for the deviation regression model method in 500,000 unrelated individuals (with parameters $p = 0.3$ and $\beta_1 = 0$). We then estimated how the power of the MZ differences approach varies for these parameters across a range of narrow-sense heritability values $\sigma^2_g = h^2$. We calculated the power for detecting a vQTL at the genome-wide significance level by drawing 1,000 replications and identifying the fraction of simulations that had $P < 5 \times 10^{-8}$.

We estimated the false discovery rate for vQTL in the presence of tagging additive loci, following the approach outlined by Hemani and colleagues[11]. Briefly, the data-generating model described above is simulated with an additive effect of $\beta_1 = 0.1$ generated, but the vQTL

test is performed at a second locus, $G^*$, that is generated to be in linkage disequilibrium with $G$. The simulations were then performed with no vQTL effect ($\beta_1 = 0$), variance of the linkage disequilibrium between $G$ and $G^*$ ($r^2 = 0,...,1$), $h^2 = 0.5$ and $p = 0.1$, and 500 repeats were performed for each parameter combination.

**GWAS model.** In our primary analysis, we estimated the association between the absolute phenotypic difference between MZ twins (residualized for age, sex and principal components and then standardized and rank transformed) and the genetic marker using linear regression for each SNP, $j$:

$$|y_{i,A} - y_{i,B}| \sim \beta_{2,j} G_{ij} + \epsilon_{i,j}.$$

We constructed two further models for sensitivity analyses: model 2, for which the within-twin mean of the phenotype was a covariate in the regression; and model 3, which differed from our primary model by not adjusting for principal components when constructing the phenotype. Model 2 was constructed to examine whether adjusting for the within-twin mean in the GWAS model would significantly impact the SNP associations, which would be the case if the MZ differences largely reflected mean differences. However, this also risks over-correcting, especially for vQTLs, which affect both the mean and variance of a phenotype, as was indicated here by a small-to-moderate ($r = -0.3–0.6$) positive correlation between the MZ phenotypic mean and MZ phenotypic differences in our sample (as was previously proposed[42]). Model 3 was constructed since some participating studies were likely to be very small (fewer than 300 participants). Including ten principal components for all studies might have been overly conservative, leading to an inflation of $P$ values.

We used the sign test to assess whether model 2 and 3 results were similar to those of model 1, as indicated by the correlation between $\beta$ values, $P$ values and the direction of the effect. The results indicated that models 2 and 3 were not significantly different from model 1 (Supplementary Table 3a–c). Therefore, we considered model 1 to be the most parsimonious, with a lower number of parameters than model 2 but similar results, while also correcting for population stratification confounding. The remaining analyses were therefore conducted using model 1 results only.

**Quality control procedure.** Study-level GWAS results (Supplementary Figs. 2–7) were quality controlled using EasyQC (version 23.8)[19]. Variants with missing estimate ($\beta$) values, standard error (s.e.) values, statistical significance ($P$) values or imputation quality score and SNPs with a minor allele frequency of <0.01 or an imputation quality score of <0.5 were removed. Cptid identifiers (chr:bp:A1:A2) were created and alleles, effects and frequencies were checked in all GWAS results and harmonized according to their respective reference panel (1000 Genomes phase 3 version 5 or the HRC). SNPs with mismatching alleles compared with the reference panel were removed. Indels, monomorphic SNPs and duplicate SNPs that could also be tri-allelic (with the same base pair position but different alleles) were removed, retaining only the SNP with the largest sample. Manhattan and QQ plots were obtained and lambda-median values were inspected for $P$ value inflation (Supplementary Figs. 2–6 and Supplementary Table 2 provide more details).

**GWAS meta-analysis.** METAL (2011 release)[20] was used to conduct inverse-variance-weighted fixed-effect meta-analysis across studies, per phenotype. First, a meta-analysis was conducted by selecting the GWAS result with the largest sample from each study, regardless of the developmental stage. Another set of GWAS meta-analyses were then conducted in developmentally stratified samples for depression (child and adult), anxiety (child and adult), ADHD (child and adolescent) and autistic traits (child and adult) phenotypes (Supplementary Table 4).

Cptid IDs were mapped into rsIDs from the 1000 Genomes phase 3 version 5 European panel. The SNP2GENE function in the FUMA web application (version 1.5.0)[21] was used to annotate GWAS SNPs and identify independent significant SNPs (SNPs in linkage disequilibrium with the lead SNP at $r^2 = 0.1$; lead SNPs are those in linkage disequilibrium with any independent significant SNPs with $r^2 > 0.6$; Supplementary Table 5) and to produce regional, QQ and Manhattan plots (Supplementary Figs. 7–11).

**Gene-based and gene set analyses.** MAGMA (version 1.08) in the FUMA web application (version 1.5.2)[21] was used to annotate GWAS SNPs and conduct gene-based and gene set analyses. Meta-analysed GWAS results were filtered to include only SNPs available in at least 50% of studies. SNPs were annotated to Ensembl version 92 protein-coding genes for gene-based analyses using default parameters (a SNP-wise model for gene testing). A competitive test was conducted for gene set analyses using default gene sets in FUMA from MsigDB version 7.0, totalling 15,496 gene sets (5,500 curated gene sets and 9,996 Gene Ontology terms). Curated gene sets contained nine data resources, including the Kyoto Encyclopedia of Genes and Genomes, Reactome and BioCarta. Gene Ontology terms comprised three categories: biological processes, cellular components and molecular functions. The major histocompatibility complex region was excluded from all annotations (Supplementary Table 5 provides details).

**Heritability analysis.** SNP heritability estimates per phenotype were obtained using LDSC (version 1.0.1)[22]. The European 1000 Genomes linkage disequilibrium scores generated by the authors of LDSC were used, and SNPs for heritability analyses were merged with a set of ~1.2 million high-quality SNPs defined by the authors of LDSC[22].

**Developmentally stratified analyses.** For ADHD symptoms, data were available for children (5–12 years of age) and adolescents (13–18 years of age), whereas for anxiety, depression and autistic traits, data were available for both children and adults (>18 years of age). The stratified analyses included a meta-analysis of GWAS results separately for samples from children, adolescents and adult for these phenotypes, followed by gene-based, gene set and heritability analyses, using the same criteria as the largest non-stratified sample.

**Mendelian randomization analysis.** We used a two-sample summary data Mendelian randomization to estimate the influence of the genetic liability of psychological phenotypes on their environmental sensitivity. Mendelian randomization uses genetic variants as instrumental variables for the exposure of interest. Three assumptions define instrumental variables: (1) relevance (the instrument must be associated with the exposure); (2) independence (there must be no uncontrolled confounders of the instrument–outcome association); and (3) exclusion restriction (the instruments only affect the outcome via the exposure of interest). We used the summary data from the MZ twins GWAS as an outcome in our Mendelian randomization analyses and investigated whether differences in the mean of each exposure (for example, depression) affect the variance in each psychiatric outcome. The exclusion restriction assumption requires that SNPs only affect the variance in the outcome via their mean effects on the exposure (liability to depression). This assumption would be violated if these variants also affected the outcome via their effects on the variance in depression, independent of their effects on the mean liability to depression. Under such a scenario, our estimates may be biased. Theoretically, this bias could be in either direction. More methodological research would be useful to determine the likelihood and magnitude of these potential biases. We selected 102 independent (linkage disequilibrium = 10,000 kilobases; $r^2 = 0.001$) genetic variants associated with depression in Howard et al.[43] as instruments for genetic liability to depression. We harmonized the effects by effect allele, chromosome and position on build hg19. We used the inverse-variance-weighted estimator to estimate the effect of genetic liability to depression on phenotypic variability using the TwoSampleMR package[44]. The variants were strongly associated with depression and there was minimal overlap between our samples and those used in the GWASs to select variants. Therefore, weak instrument bias is unlikely[45]. We followed the same procedure for the other phenotypes, but with a different number of SNPs (Supplementary Table 14). Because depression is strongly genetically correlated with anxiety but there are no well-powered GWASs for anxiety, we performed a similar analysis for variance in anxiety but using the 102 variants for depression. Here the main effects for anxiety at each of the 102 variants were obtained from a GWAS using UK Biobank data on self-reported anxiety measures[46]. Finally, we also tested educational attainment[47] and body mass index. GWAS summary statistics for main effects were obtained from OpenGWAS[48]. We reported this analysis using the STROBE-MR checklist[49].

### Reporting summary

Further information on research design is available in the Nature Portfolio Reporting Summary linked to this article.

## Data availability

Meta-analysed GWAS summary statistics from the current study are publicly available from OpenGWAS (https://gwas.mrcieu.ac.uk/). Accession codes for the GWAS meta-analyses are as follows: ieu-b-5146 (adolescent) and ieu-b-5147 (child) for MZ twin differences in ADHD symptoms; ieu-b-5148 (largest), ieu-b-5149 (adult) and ieu-b-5150 (child) for MZ twin differences in anxiety symptoms; ieu-b-5151 (largest), ieu-b-5152 (adult) and ieu-b-5153 (child) for MZ twin differences in autistic traits; ieu-b-5154 (largest), ieu-b-5155 (adult) and ieu-b-5156 (child) for MZ twin differences in depression symptoms; ieu-b-5157 for MZ twin differences in neuroticism score; ieu-b-5158 for MZ twin differences in PLEs; and ieu-b-5159 for MZ twin differences in subjective wellbeing. Data from individual studies are not publicly available and are subject to strict access control because the consent given by the participants does not allow for data storage on an individual level in repositories or journals. Access to these data requires specific approval from the relevant data access committees for each cohort. Mapping and allele frequency reference files (all based on National Center for Biotechnology Information build 37) for 1000 Genomes phase 1 version 3, 1000 Genomes phase 3 version 5 and the HRC are available via https://www.uni-regensburg.de/medizin/epidemiologie-praeventivmedizin/genetische-epidemiologie/software/index.html.

## Code availability

Scripts for GWAS analyses are available from GitHub at https://github.com/Elham-Assary/MZ-differences-GWAS.

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

## Acknowledgements

We dedicate this paper to Robert Keers, a wonderful friend, colleague, mentor and scientist, whom we lost too soon. This project continued in his absence, with contributions from many colleagues who helped to complete the work that he started. We especially thank P.B.M. for stepping in to lead the project following Rob's passing. This project was funded by: a Wellcome Trust grant (project number 208881/Z/17/Z awarded to R.K.); the National Institute for Health and Care Research (NIHR) Barts Biomedical Research Centre—a delivery partnership of Barts Health NHS Trust, Queen Mary University of London, St George's University Hospitals NHS Foundation Trust and St George's University of London (NIHR203330 awarded to P.B.M.); the Norwegian Research Council (grant number 295989), Division of Psychiatry at University College London, Medical Research Council (MRC; MR/V002147/1) and National Institute of Mental Health (MH130448) (all awarded to N.M.D.); a grant from the Wellcome Trust and Royal Society (208806/Z/17/Z) and grants from the MRC (MC_UU_00011/1 and MC_UU_00032/1 to the MRC Integrative Epidemiology Unit at the University of Bristol) (all awarded to G.H.); the Australian National Health and Medical Research Council (NHMRC) (grant APP1173025 awarded to K.L.G.); NHMRC grant APP1172990 (awarded to N.G.M.); NHMRC grant APP1172917 (awarded to S.E.M.); National Institutes of Health (NIH) grants R01AG060470, R01AG059329, R01AG050595 and R01AG046938 (awarded to C.A.R.); NIH grants R01AG04563, R01AG10175 and R01AG08861 (awarded to N.L.P.); Riksbankens Jubileumsfond (P18-0782:1) and the Swedish Research Council (2019-00244) (both awarded to S.O.); a King's International Postgraduate Research Scholarship (awarded to K.L.); the Research Council of Norway (288083) and Jacobs Foundation (2023-1510-00) (both awarded to R.C.); the European Research Council (ERC-2017-COG 771057 (WELL-BEING) awarded to MPVDW and M.B.); the Academy of Finland (grants 265240 and 263278) and Sigrid Jusélius Foundation (all awarded to J.K.); Academy of Finland grant 308698 (awarded to A.L.); Academy of Finland grant 314639 (awarded to E.V.); and the NIHR Maudsley Biomedical Research Centre (awarded to J.R.I.C.). The Danish Twin Registry has received support from the National Program for Research Infrastructure 2007 (09-063256), Danish Agency for Science, Technology and Innovation, and NIH (P01 AG08761). Genotyping at the Danish Twin Registry was conducted by the SNP&SEQ Technology Platform's Science for Life Laboratory (http://snpseq.medsci.uu.se/genotyping/snp-services/) and supported by NIH grant R01AG037985 (to N.L.P.). The Finnish Twin Cohort's phenotype and genotype data collection has received support from the Wellcome Sanger Institute, Broad Institute, European Network for Genetic and Genomic Epidemiology (FP7-HEALTH-F4-2007; grant agreement 201413), National Institute on Alcohol Abuse and Alcoholism (grants AA-12502, AA-00145 and AA-09203 to R. J. Rose, AA15416 and K02AA018755 to D. M. Dick and R01AA015416 to J. Salvatore) and Academy of Finland (grants 100499, 205585, 118555, 141054, 264146, 308248, 312073, 336823 and 352792 to J.K.). The Murcia Twin Registry is funded by the Spanish Ministerio de Ciencia, Innovación y Universidades (RTI2018-095185-B-I00), co-funded by the European Regional Development Fund. The Netherlands Twin Register is funded by the Netherlands Organisation for Scientific Research (NWO-GROOT 480-15-001/674). The OATS has received funding from an NHMRC and Australian Research Council Strategic Award Grant (part of the Ageing Well, Ageing Productively programme; 401162), NHMRC Project (seed) Grants (1024224 and 1025243), NHMRC Project Grants (1045325 and 1085606) and NHMRC Program Grants (568969 and 1093083). The OATS was facilitated through access to Twins Research Australia, a national resource supported by a Centres of Research Excellence Grant (1079102) from the NHMRC. DNA was extracted by Genetic Repositories Australia, which was funded by NHMRC Enabling Grant 401184. OATS genotyping was partly funded by a Commonwealth Scientific and Industrial Research Organisation Flagships Collaboration Fund grant. QIMR studies have received support from the following sources: the NHMRC (901061, 950998, 241944, 389875, 389891, 552485, 496682, 1009064, 552485, 496739, 1031119, 1049894, 1069141, 1086683 and 1095227), Australian Research Council (A79600334, A79801419, A79906588, DP0212016 and DP0343921), Human Frontiers Science Program (RG0154/1998-B) and Young and Well Cooperative Research Centre, which was established and funded under the Australian Government's Cooperative Research Centres programme. The Swedish Twin Registry is funded by the Swedish Research Council (2017-0064 and 421-2013-1061) and Ragnar Söderberg Foundation (E9/11). PsychChip GWAS genotyping and collaborative work at the Swedish Twin Registry was supported in part by NIH/National Institute on Aging grants R01AG037985, R01AG059329 and R01AG060470; DNA extraction was supported by grants R01AG17561 and R01AG028555; HARMONY was supported by grant R01AG08724; OCTO-Twin was supported by grant R01AG08861; gender was supported by the MacArthur Foundation Research Network on Successful Aging, Axel and Margaret Ax:son Johnson Foundation, Swedish Council for Social Research and Swedish Foundation for Health Care Sciences and Allergy Research; the Swedish Adoption/Twin Study of Aging was supported by grants R01AG04563 and R01AG10175, the MacArthur Foundation Research Network on Successful Aging, the Swedish Council For Working Life and Social Research (97:0147:1B, 2009-0795) and the Swedish Research Council (825-2007-7460 and 825-2009-6141). Research at the Swedish Twin Registry was also supported by Riksbankens Jubileumsfond (P18-0782:1) and the Swedish Research Council (2019-00244). The Twins Early Development Study is supported by a programme grant (MR/V012878/1) to T.C.E. from the UK Medical Research Council (previously MR/M021475/1 awarded to R.P.), with additional support from the NIH (AG046938). TwinsUK is funded by the Wellcome Trust, the Medical Research Council, Versus Arthritis, the European Union's Horizon 2020 programme, the Chronic Disease Research Foundation, ZOE, the NIHR Clinical Research Network and the Biomedical Research Centre based at Guy's and St Thomas' NHS Foundation Trust in partnership with King's College London. This research utilized the Queen Mary's Apocrita high-performance computing facility, supported by Queen Mary University of London's Research-IT services (https://doi.org/10.5281/zenodo.438045)[50]. The funders had no role in study design, data collection and analysis, decision to publish or preparation of the manuscript.

## Author contributions

R. Keers and E.A. were involved in conceptualizing and designing the study. R. Keers acquired funding. L.J.H., E.A., R. Keers, G.H., N.M.D. and J.K. developed the analysis pipeline. E.A., R. Keers., L.J.H., T.P., J.K. and N.M.D. were involved in data acquisition and collation. E.A. project managed and performed data analyses. J.R.I.C. and P.B.M. played a key role in supervising the analyses and P.B.M. oversaw the progress. G.H. performed simulation and Mendelian randomization analyses. E.A. wrote the first draft. E.A., J.R.I.C., P.B.M., G.H., N.M.D., J.K. and T.C.E. played a key role in interpreting the results and revising the manuscript. For the Danish Twin Registry, M.N. performed the data analysis, K.C. was the principal investigator and J.M.-F. collected and prepared the data. For the Finnish Twin Cohort, T.P. performed the data analysis, J.K. was the principal investigator and E.V. and A.L. collected and prepared the data. For the Murcia Twin Registry, J.J.M.-V. and L.C.-C. performed the data analysis and J.R.O. was the

principal investigator. For The Netherlands Twin Register, M.P.v.d.W. and J.-J.H. performed the data analysis and M.B. and D.I.B. were the principal investigators. For the OATS, A.T. and N.J.A. performed the data analysis, P. Sachdev, T.L., H.B., J.N.T., M.W. and D.A. were the principal investigators and V.S.C. and K.A.M. collected and prepared the data. For the QIMR studies, K.L.G. performed the data analysis, N. M. and S.M. were the principal investigators and S. Gordon collected and prepared the data. For the Swedish Twin Registry's ageing study, C.A.R. performed the data analysis and N.L.P. was the principal investigator. For the Swedish Twin Registry's Child and Adolescent Twin Study in Sweden, E.A. performed the data analysis, P.L. was the principal investigator and R.K., M.T., S.L. and H.L. collected and prepared the data. For the Swedish Twin Registry's PSYCH, TwinGene and Young Adult Twin Study in Sweden studies, R.A. performed the analysis and S.O. was the principal investigator. For the Twins Early Development Study, R.C. and E.A. performed the data analysis, T.C.E. and R.P. were the principal investigators and K.L. prepared the data. For TwinsUK, E.A. performed the data analysis and C.J.H. prepared the data. All authors contributed to writing and critically reviewing the manuscript.

## Competing interests

The authors declare no competing interests.

## Additional information

**Correspondence and requests for materials** should be addressed to Elham Assary.

Elham Assary [1,2] ✉, Jonathan R. I. Coleman [2,3], Gibran Hemani [4], Margot P. van de Weijer [5,6], Laurence J. Howe[4], Teemu Palviainen [7], Katrina L. Grasby [8,9,10], Rafael Ahlskog[11], Marianne Nygaard [12,13], Rosa Cheesman [14], Kai Lim[2], Chandra A. Reynolds[15,16,17], Juan R. Ordoñana [18,19], Lucia Colodro-Conde [8,18,20], Scott Gordon [8], Juan J. Madrid-Valero [18,19], Anbupalam Thalamuthu [21], Jouke-Jan Hottenga [5,6], Jonas Mengel-From [12,13], Nicola J. Armstrong [21,22], Perminder S. Sachdev [21], Teresa Lee[21], Henry Brodaty [21], Julian N. Trollor [21,23], Margaret Wright [24], David Ames[25], Vibeke S. Catts [21], Antti Latvala[26], The Within Family Consortium*, Eero Vuoksimaa[7], Travis Mallard [27,28], K. Paige Harden [29], Elliot M. Tucker-Drob [29], Sven Oskarsson [11], Christopher J. Hammond [30], Kaare Christensen [12,13,31], Mark Taylor[32], Sebastian Lundström [33,34], Henrik Larsson [32], Robert Karlsson [32], Nancy L. Pedersen[32], Karen A. Mather [21], Sarah E. Medland [8], Dorret I. Boomsma [35,36], Nicholas G. Martin [8], Robert Plomin [2], Meike Bartels [5,6], Paul Lichtenstein [32], Jaakko Kaprio [7], Thalia C. Eley [2,3,66], Neil M. Davies [37,38,66], Patricia B. Munroe [39,66] & Robert Keers[1,66,67]

[1]School of Biological and Behavioural Sciences, Queen Mary University of London, London, UK. [2]Social, Genetic and Developmental Psychiatry Centre, Institute of Psychiatry, Psychology and Neuroscience, King's College London, London, UK. [3]NIHR Maudsley Biomedical Research Centre, South London and Maudsley NHS Foundation Trust, London, UK. [4]Medical Research Council Integrative Epidemiology Unit, University of Bristol, Bristol, UK. [5]Department of Biological Psychology, Vrije Universiteit Amsterdam, Amsterdam, the Netherlands. [6]Amsterdam Public Health Institute, Amsterdam University Medical Center, Amsterdam, the Netherlands. [7]Institute for Molecular Medicine Finland, University of Helsinki, Helsinki, Finland. [8]QIMR Berghofer Medical Research Institute, Brisbane, Queensland, Australia. [9]School of Biomedical Sciences, Queensland University of Technology, Brisbane, Queensland, Australia. [10]School of Biomedical Sciences, University of Queensland, Brisbane, Queensland, Australia. [11]Department of Government, Uppsala University, Uppsala, Sweden. [12]The Danish Twin Registry, Department of Public Health, University of Southern Denmark, Odense, Denmark. [13]Department of Clinical Genetics, Odense University Hospital, Odense, Denmark. [14]The PROMENTA Research Center, Department of Psychology, University of Oslo, Oslo, Norway. [15]Institute for Behavioral Genetics, University of Colorado Boulder, Boulder, CO, USA. [16]Department of Psychology, University of California, Riverside, Riverside, CA, USA. [17]Department of Psychology and Neuroscience, University of Colorado Boulder, Boulder, CO, USA. [18]Department of Human Anatomy and Psychobiology, University of Murcia, Murcia, Spain. [19]Biomedical Research Institute of Murcia (IMIB-Arrixaca), Murcia, Spain. [20]School of Psychology, University of Queensland, Brisbane, Queensland, Australia. [21]Centre for Healthy Brain Ageing, Discipline of Psychiatry and Mental Health, School of Clinical Medicine, Faculty of Medicine and Health, UNSW Sydney, Sydney, New South Wales, Australia. [22]School of Electrical Engineering, Computing and Mathematical Sciences, Curtin University, Perth, Western Australia, Australia. [23]National Centre of Excellence in Intellectual Disability Health, Faculty of Medicine and Health, UNSW Sydney, Sydney, New South Wales, Australia. [24]Queensland Brain Institute, University of Queensland, Brisbane, Queensland, Australia. [25]University of Melbourne Academic Unit for Psychiatry of Old Age, St George's Hospital, Melbourne, Victoria, Australia. [26]Institute of Criminology and Legal Policy, University of Helsinki, Helsinki, Finland. [27]Department of Psychiatry, Harvard Medical School, Boston, MA, USA. [28]Center for Precision Psychiatry, Department of Psychiatry, Massachusetts General Hospital, Boston, MA, USA. [29]Department of Psychology and Population Research Center, University of Texas at Austin, Austin, TX, USA. [30]Department of Twin Research and Genetic Epidemiology, King's College London, London,

UK. [31]Department of Clinical Biochemistry, Odense University Hospital, Odense, Denmark. [32]Department of Medical Epidemiology and Biostatistics, Karolinska Institutet, Solna, Sweden. [33]Gillberg Neuropsychiatry Centre, University of Gothenburg, Gothenburg, Sweden. [34]Office for Psychiatry, Habilitation and Aid, Child and Adolescent Mental Health Services, Malmö, Sweden. [35]Department of Complex Trait Genetics, Center for Neurogenomics and Cognitive Research, Amsterdam Neuroscience, Vrije Universiteit Amsterdam, Amsterdam, the Netherlands. [36]Amsterdam Reproduction and Development research institute, Amsterdam, the Netherlands. [37]Division of Psychiatry, University College London, London, UK. [38]Department of Statistical Science, University College London, London, UK. [39]William Harvey Research Institute, Queen Mary University of London, London, UK. [66]These authors jointly supervised this work: Thalia C. Eley, Neil M. Davies, Patricia B. Munroe, Robert Keers. [67]Deceased: Dr. Keers. *A list of authors and their affiliations appears at the end of the paper. ✉e-mail: elham.1.assary@kcl.ac.uk

## The Within Family Consortium

Robin G. Walters[40], Sam Morris[40], Zhengming Chen[40], Kuang Lin[40], Amanda M. Hughes[4,41], Iona Y. Millwood[40], Liming Li[42,43,44], Alexandra Havdahl[45], Jean-Baptiste Pingault[2,46], W. David Hill[47], Michel Boivin[48,49], Daniel J. Benjamin[50,51,52], Matthew C. Keller[15,17], Fartein A. Torvik[53,54], Shuai Li[55,56], Eco de Geus[6,57], Floris Huider[6,57], Wonu Akingbuwa[6,57], Helga Ask[45], Per Magnus[53], Bjørn Olav Åsvold[58,59], Sonia Brescianini[60], Alexandros Giannelis[61], Emily A. Willoughby[61], Joohon Sung[62,63], Soo Ji Lee[62], Hyojin Pyun[62], David Evans[64] & Campbell Archie[65]

[40]Nuffield Department of Population Health, University of Oxford, Oxford, UK. [41]Centre for Academic Mental Health, University of Bristol, Bristol, UK. [42]Department of Epidemiology and Biostatistics, School of Public Health, Peking University, Beijing, China. [43]Peking University Center for Public Health and Epidemic Preparedness and Response, Beijing, China. [44]Key Laboratory of Epidemiology of Major Diseases (Peking University), Ministry of Education, Beijing, China. [45]PsychGen Centre for Genetic Epidemiology and Mental Health, Norwegian Institute of Public Health, Oslo, Norway. [46]Department of Clinical, Educational and Health Psychology, Division of Psychology and Language Sciences, University College London, London, UK. [47]Department of Psychology, University of Edinburgh, Edinburgh, Scotland. [48]École de Psychologie, Université Laval, Québec City, Québec, Canada. [49]Groupe de Recherche sur l'Inadaptation Psychosociale, Québec City, Québec, Canada. [50]Behavioral Decision Making Area, Anderson School of Management, University of California, Los Angeles, Los Angeles, CA, USA. [51]Human Genetics Department, David Geffen School of Medicine, University of California, Los Angeles, Los Angeles, CA, USA. [52]National Bureau of Economic Research, Cambridge, MA, USA. [53]Centre for Fertility and Health, Norwegian Institute of Public Health, Oslo, Norway. [54]The PROMENTA Research Center, University of Oslo, Oslo, Norway. [55]Centre for Epidemiology and Biostatistics, Melbourne School of Population and Global Health, University of Melbourne, Melbourne, Victoria, Australia. [56]Department of Precision Medicine, School of Clinical Sciences at Monash Health, Monash University, Melbourne, Victoria, Australia. [57]Netherlands Twin Register, Department of Biological Psychology, Vrije Universiteit Amsterdam, Amsterdam, the Netherlands. [58]HUNT Center for Molecular and Clinical Epidemiology, Department of Public Health and Nursing, Norwegian University of Science and Technology, Trondheim, Norway. [59]Department of Endocrinology, Clinic of Medicine, St. Olav's Hospital, Trondheim University Hospital, Trondheim, Norway. [60]Center for Behavioral Science and Mental Health, Istituto Superiore di Sanità, Rome, Italy. [61]Department of Psychology, University of Minnesota Twin Cities, Minneapolis, MN, USA. [62]Genome and Health Big Data Laboratory, Department of Public Health, Graduate School of Public Health, Seoul National University, Seoul, South Korea. [63]Institute of Health and Environment, Seoul National University, Seoul, South Korea. [64]Institute for Molecular Biosciences, University of Queensland, Brisbane, Australia. [65]Centre for Genomc and Experimental Medicine, Institute of Genetics & Cancer, University of Edinburgh, Edinburgh, United Kingdom.

|---|---|

# Reporting Summary

Please do not complete any field with "not applicable" or n/a.  Refer to the help text for what text to use if an item is not relevant to your study.
For final submission: please carefully check your responses for accuracy; you will not be able to make changes later.

## Statistics

For all statistical analyses, confirm that the following items are present in the figure legend, table legend, main text, or Methods section.

| n/a | Confirmed | |
|---|---|---|
| ☐ | ☑ | The exact sample size (*n*) for each experimental group/condition, given as a discrete number and unit of measurement |
| ☐ | ☑ | A statement on whether measurements were taken from distinct samples or whether the same sample was measured repeatedly |
| ☐ | ☑ | The statistical test(s) used AND whether they are one- or two-sided *Only common tests should be described solely by name; describe more complex techniques in the Methods section.* |
| ☐ | ☑ | A description of all covariates tested |
| ☐ | ☑ | A description of any assumptions or corrections, such as tests of normality and adjustment for multiple comparisons |
| ☐ | ☑ | A full description of the statistical parameters including central tendency (e.g. means) or other basic estimates (e.g. regression coefficient) AND variation (e.g. standard deviation) or associated estimates of uncertainty (e.g. confidence intervals) |
| ☐ | ☑ | For null hypothesis testing, the test statistic (e.g. *F*, *t*, *r*) with confidence intervals, effect sizes, degrees of freedom and *P* value noted *Give P values as exact values whenever suitable.* |
| ☑ | ☐ | For Bayesian analysis, information on the choice of priors and Markov chain Monte Carlo settings |
| ☑ | ☐ | For hierarchical and complex designs, identification of the appropriate level for tests and full reporting of outcomes |
| ☑ | ☐ | Estimates of effect sizes (e.g. Cohen's *d*, Pearson's *r*), indicating how they were calculated |

*Our web collection on statistics for biologists contains articles on many of the points above.*

## Software and code

Policy information about availability of computer code

| Data collection | |
|---|---|
| Data analysis | The packages used in the study are: EasyQC (v. 23.8), METAL (2011 release), MAGMA (v1.08) within FUMA web application (v1.5.2), TwoSamle MR (v.0.5.9). Scripts for GWAS analyses are available here. |

For manuscripts utilizing custom algorithms or software that are central to the research but not yet described in published literature, software must be made available to editors and reviewers. We strongly encourage code deposition in a community repository (e.g. GitHub). See the Nature Portfolio guidelines for submitting code & software for further information.

## Data

Policy information about availability of data

All manuscripts must include a data availability statement. This statement should provide the following information, where applicable:

- Accession codes, unique identifiers, or web links for publicly available datasets
- A description of any restrictions on data availability
- For clinical datasets or third party data, please ensure that the statement adheres to our policy

Meta-analysed GWAS summary statistics from the current study are publicly available from OpenGWAS (https://gwas.mrcieu.ac.uk/). **Accession codes** for GWAS meta-analyses: ADHD symptoms MZ twin differences – adolescent: ieu-b-5146, child: ieu-b-5147; Anxiety symptoms MZ twin differences – largest: ieu-b-5148, adult: ieu-b-5149, child: ieu-b-5150; Autism spectrum disorder symptoms MZ twin differences- largest: ieu-b-5151, adult: ieu-b-5152, child: ieu-b-5153; Depression symptoms MZ twin differences- largest:ieu-b-5154, adult: ieu-b-5155, child: ieu-b-5156; Neuroticism MZ twin differences score:ieu-b-5157; Psychotic-like experiences MZ twin differences: ieu-b-5158; Subjective wellbeing MZ twin differences: ieu-b-5159.
Data from individual studies are not publicly available and are subject to strict access control, since the consent given by the participants does not allow for data storage on an individual level in repositories or journals. Access to these data requires specific approval from the relevant data access committees for each cohort.

Mapping and allele frequency reference files (all based on NCBI build 37) for 1000G phase1 version3, 1000G phase3 version5, and Haplotype Reference Consortium (HRC) are

available via https://www.uni-regensburg.de/medizin/epidemiologie-praeventivmedizin/genetische-epidemiologie/software/index.html.

## Research involving human participants, their data, or biological material

Policy information about studies with [human participants or human data](). See also policy information about [sex, gender (identity/presentation), and sexual orientation]() and [race, ethnicity and racism]().

| | |
|---|---|
| Reporting on sex and gender | The study reports total sample sizes as well as the number of females in the data used in the analyses. Data on sex were obtained via self-reports, parent reports (for children) and birth registry. No sex or gender-based analyses have been conducted due to low power. Findings do not apply to a specific sex or gender. |
| Reporting on race, ethnicity, or other socially relevant groupings | The study does not include reporting on ethnicity/race or other socially relevant groupings |
| Population characteristics | The sample consisted exclusively of monozygotic twins, registered in twin cohorts |
| Recruitment | The study did not involve participant recruitment as the data for analysis was already collected via participating twin cohorts. Different recruitment methods were used in each cohort, including recruitment through birth registries, hospital records and national registers. More details on recruitment for each participating cohort is provided in supplementary materials. |
| Ethics oversight | Ethical approval was obtained separately in each participating study. The details are presented in the manuscript under methods section. |

Note that full information on the approval of the study protocol must also be provided in the manuscript.

# Field-specific reporting

Please select the one below that is the best fit for your research. If you are not sure, read the appropriate sections before making your selection.

☐ Life sciences ☑ Behavioural & social sciences ☐ Ecological, evolutionary & environmental sciences

For a reference copy of the document with all sections, see [nature.com/documents/nr-reporting-summary-flat.pdf]()

# Life sciences study design

All studies must disclose on these points even when the disclosure is negative.

| | |
|---|---|
| Sample size | |
| Data exclusions | |
| Replication | |
| Randomization | |
| Blinding | |

# Behavioural & social sciences study design

All studies must disclose on these points even when the disclosure is negative.

| | |
|---|---|
| Study description | This was a genome-wide association study of quantitative data on seven psychological traits (attention deficit hyperactivity symptoms, autistic traits, anxiety and depression symptoms, psychotic-like experiences, neuroticism, and wellbeing), assessed in monozygotic twins along with genome-wide genetic data |
| Research sample | The sample consisted of monozygotic twins, with participants' age ranging between 7 to 70 years old across studies, with approximately half of the sample reported as being female. Mean age across studies, number of females per phenotype and total number of participants in the study are presented in Table 1 and per cohort information are presented in SI. |
| Sampling strategy | Stratified sampling procedure: analysis of existing data only from monozygotic twins registered in twin cohorts. The planned sample size was to collate the largest possible data on MZ twins with DNA data from existing datasets. The final sample size was determined by availability of genome-wide genetic and phenotypic data in twin cohorts who contributed to the study. |
| Data collection | No data was collected for the purpose of this study as it involved secondary data analysis of existing datasets. The data collection method varied by participating twin cohorts, including interviews, pen and paper questionnaires and online questionnaires. |
| Timing | Secondary analysis of GWAS data from participating cohorts were carried out between September 2020 to February 2022. |
| Data exclusions | Samples with genetic data missingness more than 5% and genetic data missing in more than 95% of the sample were excluded from analyses. Data that were available only for one twin from each pair were excluded, as the phenotype construction step (phenotype differences) requires data from both twins, as well as individuals with missing information on covariates such as sex and age to account for demographic effects |
| Non-participation | No participants were involved in the study, as this was secondary data analysis of existing datasets. |
| Randomization | Randomisation did not apply to this study design, as it was a correlational study of genetic and phenotypic data |

# Ecological, evolutionary & environmental sciences study design

All studies must disclose on these points even when the disclosure is negative.

| | |
|---|---|
| Study description | |
| Research sample | |
| Sampling strategy | |
| Data collection | |
| Timing and spatial scale | |
| Data exclusions | |
| Reproducibility | |
| Randomization | |
| Blinding | |

Did the study involve field work?   ☐ Yes   ☐ No

## Field work, collection and transport

| | |
|---|---|
| Field conditions | |
| Location | |
| Access & import/export | |
| Disturbance | |

# Reporting for specific materials, systems and methods

We require information from authors about some types of materials, experimental systems and methods used in many studies. Here, indicate whether each material, system or method listed is relevant to your study. If you are not sure if a list item applies to your research, read the appropriate section before selecting a response.

## Materials & experimental systems

| n/a | Involved in the study |
|---|---|
| ☑ | ☐ Antibodies |
| ☑ | ☐ Eukaryotic cell lines |
| ☑ | ☐ Palaeontology and archaeology |
| ☑ | ☐ Animals and other organisms |
| ☑ | ☐ Clinical data |
| ☑ | ☐ Dual use research of concern |
| ☑ | ☐ |

## Methods

| n/a | Involved in the study |
|---|---|
| ☑ | ☐ ChIP-seq |
| ☑ | ☐ Flow cytometry |
| ☑ | ☐ MRI-based neuroimaging |

## Antibodies

| | |
|---|---|
| Antibodies used | |
| Validation | |

# Eukaryotic cell lines

Policy information about cell lines and Sex and Gender in Research

| | |
|---|---|
| Cell line source(s) | |
| Authentication | |
| Mycoplasma contamination | |
| Commonly misidentified lines (See ICLAC register) | |

# Palaeontology and Archaeology

| | |
|---|---|
| Specimen provenance | |
| Specimen deposition | |
| Dating methods | |

☐ Tick this box to confirm that the raw and calibrated dates are available in the paper or in Supplementary Information.

| | |
|---|---|
| Ethics oversight | |

Note that full information on the approval of the study protocol must also be provided in the manuscript.

# Animals and other research organisms

Policy information about studies involving animals; ARRIVE guidelines recommended for reporting animal research, and Sex and Gender in Research

| | |
|---|---|
| Laboratory animals | |
| Wild animals | |
| Reporting on sex | |
| Field-collected samples | |
| Ethics oversight | |

Note that full information on the approval of the study protocol must also be provided in the manuscript.

# Clinical data

Policy information about clinical studies
All manuscripts should comply with the ICMJE guidelines for publication of clinical research and a completed CONSORT checklist must be included with all submissions.

| | |
|---|---|
| Clinical trial registration | |
| Study protocol | |
| Data collection | |
| Outcomes | |

# Dual use research of concern

Policy information about dual use research of concern

## Hazards

Could the accidental, deliberate or reckless misuse of agents or technologies generated in the work, or the application of information presented in the manuscript, pose a threat to:

| No | Yes | |
|----|-----|---|
| ☐ | ☐ | Public health |
| ☐ | ☐ | National security |
| ☐ | ☐ | Crops and/or livestock |
| ☐ | ☐ | Ecosystems |
| ☐ | ☐ | Any other significant area |

## Experiments of concern

Does the work involve any of these experiments of concern:

| No | Yes | |
|----|-----|---|
| ☐ | ☐ | Demonstrate how to render a vaccine ineffective |
| ☐ | ☐ | Confer resistance to therapeutically useful antibiotics or antiviral agents |
| ☐ | ☐ | Enhance the virulence of a pathogen or render a nonpathogen virulent |
| ☐ | ☐ | Increase transmissibility of a pathogen |
| ☐ | ☐ | Alter the host range of a pathogen |
| ☐ | ☐ | Enable evasion of diagnostic/detection modalities |
| ☐ | ☐ | Enable the weaponization of a biological agent or toxin |
| ☐ | ☐ | Any other potentially harmful combination of experiments and agents |

# Plants

Seed stocks

Novel plant genotypes

Authentication

# ChIP-seq

## Data deposition

☐ Confirm that both raw and final processed data have been deposited in a public database such as GEO.

☐ Confirm that you have deposited or provided access to graph files (e.g. BED files) for the called peaks.

Data access links
*May remain private before publication.*

Files in database submission

Genome browser session
(e.g. UCSC)

## Methodology

Replicates

Sequencing depth

Antibodies

Peak calling parameters

Data quality

| Software | |
|---|---|

# Flow Cytometry

## Plots

Confirm that:

☐ The axis labels state the marker and fluorochrome used (e.g. CD4-FITC).

☐ The axis scales are clearly visible. Include numbers along axes only for bottom left plot of group (a 'group' is an analysis of identical markers).

☐ All plots are contour plots with outliers or pseudocolor plots.

☐ A numerical value for number of cells or percentage (with statistics) is provided.

## Methodology

| Sample preparation | |
|---|---|
| Instrument | |
| Software | |
| Cell population abundance | |
| Gating strategy | |

☐ Tick this box to confirm that a figure exemplifying the gating strategy is provided in the Supplementary Information.

# Magnetic resonance imaging

## Experimental design

| Design type | |
|---|---|
| Design specifications | |
| Behavioral performance measures | |

| Imaging type(s) | |
|---|---|
| Field strength | |
| Sequence & imaging parameters | |
| Area of acquisition | |

Diffusion MRI    ☐ Used    ☐ Not used

## Preprocessing

| Preprocessing software | |
|---|---|
| Normalization | |
| Normalization template | |
| Noise and artifact removal | |
| Volume censoring | |

## Statistical modeling & inference

| Model type and settings | |
|---|---|
| Effect(s) tested | |

Specify type of analysis: ☐ Whole brain ☐ ROI-based ☐ Both

Statistic type for inference

[                    ]

(See Eklund et al. 2016)

Correction

[                    ]

## Models & analysis

| n/a | Involved in the study |
|-----|------------------------|
| ☐ ☐ | Functional and/or effective connectivity |
| ☐ ☐ | Graph analysis |
| ☐ ☐ | Multivariate modeling or predictive analysis |

Functional and/or effective connectivity [                    ]

Graph analysis [                    ]

Multivariate modeling and predictive analysis [                    ]

