## [Peer Review File · Nature Human Behaviour]

Genetics of monozygotic twins reveals the impact of environmental sensitivity on psychiatric and neurodevelopmental phenotypes

Corresponding Author: Dr Elham Assary

Version 0:

Decision Letter:

15th August 2024

Dear Dr Assary,

Thank you once again for your manuscript, entitled "Genetics of environmental sensitivity to psychiatric and neurodevelopmental phenotypes: evidence from GWAS of monozygotic twins", and for your patience during the peer review process.

Your Article has now been evaluated by 2 referees. Please note that we had invited an additional reviewer, but they have failed to return their comments. You will see from the comments copied below that, although the reviewers find your work of potential interest, they have raised quite substantial concerns. In light of these comments, we cannot accept the manuscript for publication, but would be interested in considering a revised version if you are willing and able to fully address reviewer and editorial concerns.

We hope you will find the referees' comments useful as you decide how to proceed. If you wish to submit a substantially revised manuscript, please bear in mind that we will be reluctant to approach the referees again in the absence of major revisions. We are committed to providing a fair and constructive peer-review process. Do not hesitate to contact us if there are specific requests from the reviewers that you believe are technically impossible or unlikely to yield a meaningful outcome.

In particular, although both reviewers appreciate the methodological novelty of your work, Reviewer #3 is particularly concerned about very limited power. Both reviewers recommend additional analyses that will strengthen the work. We ask that you perform these. However, we understand that the issue of power cannot be resolved. For this reason, we ask that you make this limitation very salient in the manuscript (including in the abstract) and focus primarily on the methodological advance the work represents, with caveated discussion of the results.

If you wish to submit a suitably revised manuscript, we would hope to receive it within 4 months. I would be grateful if you could contact us as soon as possible if you foresee difficulties with meeting this target resubmission date.

- Include a "Response to the editors and reviewers" document detailing, point-by-point, how you addressed each editor and referee comment. If no action was taken to address a point, you must provide a compelling argument. When formatting this document, please respond to each reviewer comment individually, including the full text of the reviewer comment verbatim followed by your response to the individual point. This response will be used by the editors to evaluate your revision and sent back to the reviewers along with the revised manuscript.
- Highlight all changes made to your manuscript or provide us with a version that tracks changes.

Link Redacted

Thank you for the opportunity to review your work. Please do not hesitate to contact me if you have any questions or would like to discuss the required revisions further.

Sincerely,

[Redacted Signature]

Nature Human Behaviour

Reviewer expertise:

Reviewer #1: Gene x Environment interactions, GWAS, MZ twins

Reviewer #3: Gene x Environment interactions, GWAS

REVIEWER COMMENTS:

Reviewer #1:

Remarks to the Author:

The manuscript entitled "Genetics of environmental sensitivity to psychiatric and neurodevelopmental phenotypes: evidence from GWAS of monozygotic twins" investigates genetic influences on environmental sensitivity by conducting a large-scale GWAS meta-analysis on monozygotic (MZ) twins. Using data from 21,792 MZ twins across 11 studies, the research explores seven psychiatric and neurodevelopmental phenotypes, including ADHD, autistic traits, anxiety, depression, psychotic-like experiences, neuroticism, and wellbeing. The analysis identifies 13 genome-wide significant associations, highlighting genes related to stress-reactivity, growth factors, and catecholamine uptake. Despite imprecise SNP-heritability estimates, findings suggest genetic contributions to environmental sensitivity. This MZ twin-based approach mitigates biases found in population-based methods, offering robust insights into genotype-environment interactions and phenotypic variance.

Strengths:

There are several strengths of that makes this study really unique: 1) it utilizes the unique feature of twin study: The use of monozygotic twins to investigate genetic contributions to environmental sensitivity is a novel approach that mitigates many biases present in population-based studies. This method provides more accurate estimates of genetic influence by controlling for genetic background and familiar environmental factors. 2) large sample size: the study combines many twin studies and integrates into a meta-analysis which significantly increases the statistical power. 3) Rigorous statistical analysis: the manuscript employs various additional statistical methods, including gene-based, gene-set analyses, as well as Mendelian randomization. This multifaceted approach strengthens the validity of the findings.

Weakness:

The authors have explicitly points out the weakness of this study regarding: 1) Generalizability: The sample is exclusively of European ancestry, which raises concerns about the generalizability of the findings to other populations. Future studies should aim to include more diverse populations to validate and extend these findings. 2) The reliance on MZ twin differences to reflect genuine phenotypic variability might be confounded by measurement errors. Additionally, it is unclear if these differences are stable over time, which could impact the reliability of the associations found. If a longitudinal study using such method can be also performed, the results could be more convincing.

Major Concerns:

There are several concerns and confusions when I was reading this manuscript: 1) I get that the vQTL analysis is the method to find genetic loci which mostly affect the variance of certain phenotype. But I still have a hard time understanding how this relates to environmental sensitivity. Maybe the authors should put a little more introduction for the reader to fully understand this topic and why finding vQTL is important and how it is mathematically related with QTL, and whether there are a large number of QTL overlapping with vQTL. 2) how this results compare to the population level results? Even though population-based results might be inflated or biased in some sense, they still have much higher statistical power due to a much larger sample size, I wonder how the two different types of results compare to each other. I also wonder whether their differences could be identified by simulations. 3) I am also interested in whether there is a gender-specific vQTL. It is very intriguing to know how male and female differ in terms of vQTL. Any additional analysis considering this topic will be much appreciated. Some final thoughts: I think the nice thing about the twin study is we usually collect both MZ and DZ twins, this study only utilizes MZ because the phenotypic difference between MZ twins can get rid of subtract the genetic and shared environmental effect. I wonder if there is any way to also integrate the difference of MZ into this vQTL framework since I still believe there has to be some useful information to contribute to this twin-based vQTL analysis. Even though I still cannot figure out how, it is still worth considering for future studies.

Overall, this manuscript presents a significant advancement in understanding the genetics of environmental sensitivity using a robust methodological framework, though there remain areas for further refinement and exploration.

Reviewer #3:

Remarks to the Author:

The paper represent a massive collaboration among several twin data sets to achieve an impressive sample size of twins used to estimate a GWAS of within-MZ-twin differences in psychiatric and neurodevelopmental phenotypes.

The idea of using identical twins to understand phenotypic differences and environmental sensitivity has strong theoretical foundations in the literature. The analysis seems to me extremely rigorous, pre-registered, well executed, and at the frontier of the field.

In theory, the paper represent a frontier execution of an important idea.

In practice, however, it seems to me that the sample size is still too small to achieve enough statistical power in order to learn something meaningful. This is according to the author's own simulations. I don't see any practical and feasible way to overcome this problem.

I still provide a few comments and suggestions hoping that they can be helpful.

- Siblings: why not using also fraternal twins and possibly siblings in order to increase the sample size? The simulations in the paper seem to accept an additive genetic model, with no gene-gene interactions, suggesting that the theoretical added value of identical twins might not justify the reduction in sample size
- non-Europeans: why was there a restriction on genetic ancestry? Having a within family design controls for population stratification. Including other ancestries in genetically informed studies has also been advocated by the literature. Including the Texas twin study, for example, seems easily within the scope of the current set of authors.
- genetic correlations: would it be possible to show some genetic correlations between the results of this paper's GWAS summary statistics and other (mean or variance) GWAS? For example the mean GWAS for the same traits, or the population-based variance GWAS, etc.

On page 5 the authors mentioned that they "could not estimate the genetic correlation between all phenotypes because the SNP-heritabilities were too low and imprecise" Is that true of genetic correlations with other well-powered GWAS as well? I would find those results much more informative than the two-sample MR.

Smaller comments:

- mendelian randomization: I'm not sure the current setting satisfies the exclusion restriction assumption needed to perform MR. If I understand correctly, one would need to assume that certain genetic variants only influence directly the mean but only indirectly the variance of a phenotype, while actually testing that the mean of a phenotype has a causal effect on its variance. I would drop the MR analysis, or at least warrant the reader about the plausibility of the necessary assumptions in the current setting.
- simulations: a few questions on the simulation
 - given the paper's estimates of heritability for the phenotypes in question, it seems to me that figure 5 suggests that a population based design with 20k individuals actually has more power. In other words, it might give more reliable results to use the MZ twins as if they were just random people instead of twins. Is that correct? if so, I think it should be pointed out somewhere in the text.
 - instead of/decide that current graph in figure 5 I would suggest adding another one where heritability is fixed at the level estimated for the current phenotypes and what varies on the x-axis is the simple size of twins needed to achieve a certain statistical power. This would help future researchers who might want to replicate the current results once more twin data is available.
 - in the method section, I don't fully understand the notation of the equations on page 9. Shouldn't G_i also be indexed by the SNP, G_{ij} ? Shouldn't there be exclamation over different j SNP somewhere in the data generating model of the phenotype y ? Why is the mean of that $z_i = 0$, aren't there any average effects of the remaining polygenic risk? What is ϵ_i ? Is $\sigma^2_v = \beta_2 G_i$?

Version 1:

Decision Letter:

Our ref: NATHUMBEHAV-24041546A

3rd February 2025

Dear Dr Assary,

Thank you for submitting your revised manuscript "Genetics of environmental sensitivity to psychiatric and neurodevelopmental phenotypes: evidence from GWAS of monozygotic twins" (NATHUMBEHAV-24041546A). It has now been seen by the original referees and their comments are below. As you can see, the reviewers find that the paper has improved in revision. We will therefore be happy in principle to publish it in Nature Human Behaviour, pending minor revisions to satisfy the referees' final requests and to comply with our editorial and formatting guidelines.

We are now performing detailed checks on your paper and will send you a checklist detailing our editorial and formatting requirements within two weeks. Please do not upload the final materials and make any revisions until you receive this additional information from us.

Sincerely,

[Redacted signature]

Nature Human Behaviour

Reviewer #3 (Remarks to the Author):

I appreciate the authors' thoughtful and detailed responses to my comments. The revisions have improved the manuscript significantly, and I find that almost all of my concerns have been fully addressed. The authors have provided additional clarifications and modifications that strengthen the interpretation of their results. Their discussion of statistical power, the feasibility of integrating dizygotic twins and siblings, and the challenges of increasing sample sizes for this kind of study is particularly useful. I also acknowledge their efforts to incorporate discussion on genetic correlations and the challenges of including non-European ancestry samples.

However, I remain unconvinced by the Mendelian Randomization (MR) analysis. First of all, repeating the assumptions in line 477 and 478 is unnecessary: they are already clearly stated just above, in line 471-474. The point is not stating the assumptions, but arguing about their plausibility in this particular setting.

My concern lies in the plausibility of the exclusion restriction assumption. The authors state that this assumption is plausible and reference the STROBE-MR checklist. However, in this specific context, the assumption that certain SNPs affecting the mean of a trait cannot directly influence its variance seems particularly strong. Given the biological complexity of genetic effects, it is difficult to justify why such SNPs would not have any direct impact on variance beyond their influence through the mean.

To my knowledge, there are no empirical methods to directly test the exclusion restriction assumption in this setting, making it an issue of interpretation. I would appreciate further discussion from the authors on why they believe this assumption holds in their analysis, or what might happen if this is not the case in practice: would they expect a positive or negative bias in their causal analysis?

More generally, is it that important in this setting to claim causality? What do we learn from this MR analysis, what are the consequences of these results for researchers and practitioners? The results suggest that being depressed increases phenotypic variability of traits: does that mean that treating depression might reduce the variability of these traits? Would this improve our understanding of depression? How? I believe there's a lot in the paper already, and maybe simply showing the effect of the top-SNPs of Howard et al. on phenotypic variability would spark enough future research and discussion.

In any case, this is just a minor point in an already long and beautiful paper. I have already pushed too much. I think the authors should decide on their own what to do with this MR analysis, which is certainly done rigorously and precisely.

Overall, the manuscript presents a rigorous and important contribution, but I encourage the authors to carefully reconsider or at least further justify the MR analysis and its underlying assumptions.

Institute of Psychiatry,
Psychology & Neuroscience
Social, Genetic & Developmental
Psychiatry Centre

10th December 2024

Ref: NATHUMBEHAV-24041546-T

Dear reviewers,

Thank you for taking the time to read our manuscript titled "**Genetics of environmental sensitivity to psychiatric and neurodevelopmental phenotypes: evidence from GWAS of monozygotic twins**" and for your thoughtful and constructive feedback.

We have carefully considered your comments and made amendments to our manuscript accordingly. We believe the paper is strengthened as a result.

Thank you for considering our revised manuscript. We hope that we have satisfactorily addressed all the points that have been raised.

Best wishes,

Dr Elham Assary, on behalf of the authors
Social, Genetic and Developmental Psychiatry Centre,
Institute of Psychiatry, Psychology and Neuroscience,
King's College London, London, UK
Email: elham.1.assary@kcl.ac.uk

Reviewers' comments:

Reviewer #1:

1- The manuscript entitled "Genetics of environmental sensitivity to psychiatric and neurodevelopmental phenotypes: evidence from GWAS of monozygotic twins" investigates genetic influences on environmental sensitivity by conducting a large-scale GWAS meta-analysis on monozygotic (MZ) twins. Using data from 21,792 MZ twins across 11 studies, the research explores seven psychiatric and neurodevelopmental phenotypes, including ADHD, autistic traits, anxiety, depression, psychotic-like experiences, neuroticism, and wellbeing. The analysis identifies 13 genome-wide significant associations, highlighting genes related to stress-reactivity, growth factors, and catecholamine uptake. Despite imprecise SNP-heritability estimates, findings suggest genetic contributions to environmental sensitivity. This MZ twin-based approach mitigates biases found in population-based methods, offering robust insights into genotype-environment interactions and phenotypic variance.

Strengths: There are several strengths of that makes this study really unique: 1) it utilizes the unique feature of twin study: The use of monozygotic twins to investigate genetic contributions to environmental sensitivity is a novel approach that mitigates many biases present in population-based studies. This method provides more accurate estimates of genetic influence by controlling for genetic background and familiar environmental factors. 2) large sample size: the study combines many twin studies and integrates into a meta-analysis which significantly increases the statistical power. 3) Rigorous statistical analysis: the manuscript employs various additional statistical methods, including gene-based, gene-set analyses, as well as Mendelian randomization. This multifaceted approach strengthens the validity of the findings.

A: Many thanks for your positive review of our study and identifying the main strengths of the paper.

2 - There are several concerns and confusions when I was reading this manuscript:

- I get that the vQTL analysis is the method to find genetic loci which mostly affect the variance of certain phenotype. But I still have a hard time understanding how this relates to environmental sensitivity. Maybe the authors should put a little more introduction for the reader to fully understand this topic and why finding vQTL is important and how it is mathematically related with QTL, and whether there are a large number of QTL overlapping with vQTL.

A: We have now extended the introduction on this topic and improved the rationale for this approach. We now write on pages 3 & 4: "Interactions between genetic variants and the environment increase phenotypic variability, which may be reflected in differences in mean and/or variance of a phenotype in a population. This is evidenced when a genotype is associated with phenotype levels but only in certain environmental conditions. Environmental sensitivity can also increase the variance of a trait if a genotype produces a wide range of phenotypes depending on environmental exposures, which may or may not also affect its population mean. Genetic knowledge of environmental sensitivity is most consistently exploited in bioengineering and evidenced in behavioural ecology, but it has been extremely challenging to evaluate in humans, especially for psychiatric disorders (Assary et al., 2018). Understanding the genetic basis of environmental sensitivity is crucial for improving human health, as it informs on the biological pathways implicated in variations in response to environmental exposures. [...] (vQTL) analysis [...] aims to discover genetic variants associated with phenotypic variance, which can be prioritised for a statistical test of

gene-environment interaction ¹¹. [...] it is therefore challenging to robustly determine which potential mechanisms have given rise to the observed trait variance associated with a vQTL.

[...] While statistical correction for certain observed demographic effects (e.g., age, sex) is possible, they cannot be corrected for unobserved confounders (e.g., residual population stratification, dynastic effects via parents, assortative mating).

[...] MZ twin pairs have the same degree of genetic similarity [...]. The greater within-pair differences, in a population of MZ twins, therefore, reflects the pairs' greater sensitivity to their non-shared environments. The greater within-pair differences, in a population of MZ twins, therefore, reflects the pairs' greater sensitivity to their non-shared environments. Jinks and Fulker ¹⁶ proposed a test in monozygotic twins which involved obtaining the association of the MZ pair differences with the MZ-pair mean score and they provided a full description of its standard biometrical terms. The rationale of their test of genotype-environment interaction is the same one which underlies tests involving inbred lines of animals, which detect genotype-environment interaction through heterogeneity of within-strain variances. We may now take SNPs as measured genotype indicators and test for associations with within-pair pair differences.

A GWAS of MZ phenotypic differences can identify the loci associated with variations in environmental sensitivity while also accounting for dynastic and epistasis effects, which are difficult to account for in population-based approaches."

- how this results compare to the population level results? Even though population-based results might be inflated or biased in some sense, they still have much higher statistical power due to a much larger sample size, I wonder how the two different types of results compare to each other. I also wonder whether their differences could be identified by simulations.

A: Ideally, we would be able to compare the vQTL estimates from population studies with our MZ-derived estimate. However, there are currently no population-based vQTL studies of our psychiatric phenotypes. This is why we instead used MR analyses that showed the top SNPs associated with increased mean levels of phenotypes, such as depression, found in GWAS of population-level data, were associated with variance in our sample. Figure 5 shows the results of simulations, which suggest the MZ method is more powerful than the population-based vQTL analyses when narrow-sense heritability is at least moderate. Visscher & Posthuma, 2010, also compare the power for detecting environmental sensitivity using MZ vs population-based approach.

- 3- I am also interested in whether there is a gender-specific vQTL. It is very intriguing to know how male and female differ in terms of vQTL. Any additional analysis considering this topic will be much appreciated.

A: At the outset, we asked participating cohorts to run their GWAS separately stratified by male and female twins. We decided not to analyse or present the stratified data for two main reasons: 1) the sample sizes were inevitably much smaller than the total sample, with many of the phenotypes having male/female samples of < 5k, so the stratified sample has far lower power than the total sample, 2) presenting the results of the parallel sex-stratified samples and interpreting the findings would make an already packed paper into an unwieldy one.

- 4- Some final thoughts: I think the nice thing about the twin study is we usually collect both MZ and DZ twins, this study only utilizes MZ because the phenotypic difference between MZ twins can

get rid of subtract the genetic and shared environmental effect. I wonder if there is any way to also integrate the difference of MZ into this vQTL framework since I still believe there has to be some useful information to contribute to this twin-based vQTL analysis. Even though I still cannot figure out how, it is still worth considering for future studies.

A: We presume you mean DZ twins in the second sentence? We also think integrating DZ twin data into vQTL analysis should be possible. There are two technical challenges to integrating DZ data for vQTL estimation. First, though DZ twins will have pseudo-MZ regions that are shared IBD at both loci, those regions may be in LD with non-IBD regions with mean effects. We are currently developing methodological advances to handle this issue, which is substantial separate work. Secondly, a more minor issue is that it will not necessarily control for epistasis. While DZ twins may be similar at specific points in the genome, they will differ at other points. So, vQTLs identified by DZs may also reflect these epistatic effects. However, an advantage of DZ (and genotyped siblings) is that there are larger samples of DZs and siblings, which should have greater power for detecting vQTLs, but still account for demographic effects and other forms of environmental confounding. The lead author is currently securing funding for a project incorporating DZ twins' data to identify vQTLs, which also allows for comparison with MZ differences estimates to determine the contribution of various mechanisms to phenotypic variation. Finally, on a practical basis, this project is out of scope for the current paper as the analysis we undertook with the MZs took five years to complete.

- 5- Overall, this manuscript presents a significant advancement in understanding the genetics of environmental sensitivity using a robust methodological framework, though there remain areas for further refinement and exploration.

A: Many thanks for your positive assessment of the overall contribution of this paper to the field. Your review has improved the paper.

Reviewer #3:

- 1- The paper represents a massive collaboration among several twin data sets to achieve an impressive sample size of twins used to estimate a GWAS of within-MZ-twin differences in psychiatric and neurodevelopmental phenotypes. The idea of using identical twins to understand phenotypic differences and environmental sensitivity has strong theoretical foundations in the literature. The analysis seems to me extremely rigorous, pre-registered, well executed, and at the frontier of the field. In theory, the paper represents a frontier execution of an important idea. In practice, however, it seems to me that the sample size is still too small to achieve enough statistical power in order to learn something meaningful. This is according to the author's own simulations. I don't see any practical and feasible way to overcome this problem.

A: We agree that MZ twins provide one of the most robust sources of evidence about environmental sensitivity and that the biggest challenge to this approach is sample size. Here, we have analysed data from an order of magnitude more samples than has been in previous studies. This was only made possible through collaboration with twin cohorts and their analysts and PIs, who have kindly shared their summary results. Hence, our sample provides some of the most robust and precise evidence to date about environmental sensitivity. Our paper makes two important contributions to advancing knowledge of environmental sensitivity: 1) showing the potential value of twin cohorts in facilitating research and genetic discovery and encouraging funding and recruitment of such cohorts. This ensures future studies on this topic can be run in larger sample sizes that are better powered, and therefore, findings are more robust. There are currently over 1.3 million MZ and DZ twins in various registries in the world (Hur et al., 2019),

but only a minority have been genotyped. The power of this approach is likely to be addressed in the future by genotyping these cohorts; 2) stimulating future research by highlighting alternative quasi-experimental methodologies as proposed here to address some of the limitations of other approaches for investigating the genetics of environmental sensitivity. Thus far, psychiatric genetics' research methods have **exclusively** focused on statistical tests of gene-environment interactions to address genetic sensitivity to the environment (Assary et al. 2018), which have been inconsistent and unreliable (i.e. candidate GxE studies). This paper provides an alternative, complementary source of evidence, allowing us to triangulate conclusions across multiple approaches. Our findings and future efforts using these methods will inform on the genetic basis of environmental sensitivity. Finally, we point out that genome-wide association studies of complex traits started with the WTC that included a few thousand cases in 2007 (in Nature) and were massively underpowered; 17 years later, as of 2024-11-20, the GWAS Catalog (<https://www.ebi.ac.uk/gwas/>) contains 7083 publications, 692444 top associations -without accounting for LD. Over time, the power and precision of MZ GWAS will only increase. However, we need studies such as this one to demonstrate the feasibility of this approach and stimulate advances in the field. We have now highlighted the current power limitations in the abstract, the manuscript, and the discussion of the results.

In the abstract, we write: "This is the largest genetic study of Monozygotic twins to date by an order of magnitude, evidencing an alternative method to study the genetics of environmental sensitivity. Statistical power was limited for some analyses, which calls for better-powered future studies to enable further insights into the genetic architecture of environmental sensitivity."

On page 5, in the results section, we write: "Overall, despite this being the largest sample of MZ twins to date, low power resulted in large confidence intervals around the heritability estimates."

In the discussion section on page 6, we write: "The main limitation was limited statistical power to detect small genetic effects on variance. The findings should, therefore, be interpreted in light of our statistical power".

Siblings: why not using also fraternal twins and possibly siblings in order to increase the sample size? The simulations in the paper seem to accept an additive genetic model, with no gene-gene interactions, suggesting that the theoretical added value of identical twins might not justify the reduction in sample size

A: This is an important future direction that we are pursuing. However, there are two technical challenges. First, though DZ twins will have pseudo-MZ regions that are shared IBD at both loci, those regions may be in LD with non-IBD regions with mean effects. We are developing methodological advances to handle this issue that can induce substantial bias in vQTL estimates, but this is a substantial separate piece of work. Second, the simulations are based on MZ data, so while g_xg interactions may be present, they will be the same for both twins in a pair, so don't contribute to the difference between them. However, this would not be the case in sibling/DZ data. DZ twins/siblings are different due to differences in their genes and environments, so epistasis effects could play a significant role in the observed phenotypic differences with their sibling/co-twin, whereas, for a population of MZ twins, the only source of difference with their co-twin is due to their non-shared environments (no gene-gene interactions and epistasis can be excluded as an explanation). If we were interested in phenotypic variation, regardless of the underlying mechanisms that give rise to the observed

variations (GxG, GxE, G-E correlation), using DZ or siblings would be recommended. However, MZ twins provide the most robust evidence of GxE because only this design can exclude GxG. The lead author is looking to secure funding to examine vQTLs using DZ populations and sibs, which is preferable to unrelated samples since it accounts for demographic and other effects that are hard to account for in unrelated samples.

- 2- non-Europeans: why was there a restriction on genetic ancestry? Having a within family design controls for population stratification. Including other ancestries in genetically informed studies has also been advocated by the literature. Including the Texas twin study, for example, seems easily within the scope of the current set of authors.

A: Thank you for your question on this. We agree that research must consider and incorporate multiple ancestries in genetic research. At the time of designing the study (mid-2019), we searched and contacted twin cohorts across the world to identify which ones also collected genome-wide DNA data so we could invite them to take part in our study. To our knowledge, no such dataset existed in Africa or Latin America, and we could not ascertain the participation of the only cohort in Asia that appeared to have genome-wide data. For example, COTASS twin study in Sri Lanka has biological material, but no funding or infrastructure for genotyping (Dissanayake et al., 2024). So, although there are many twin cohorts worldwide, few have genome-wide DNA data; such studies that do are generally based in USA/Europe/Australia. Since the individual cohorts in these countries were already relatively small (100 – 3,000), it was deemed unlikely to have enough samples (e.g., less than 100 in most country-representative cohorts) to conduct the studies separately for different ancestries within cohorts. We invited Texas Twins to participate, and they contributed to the study. Still, the small initial sample of twins with complete twin data was 100, which was further reduced following the QC process, such as removing per SNP sample missingness and low MAF, resulting in highly skewed data, so it was not included in meta-analyses (information contained in supplementary materials). We had already noted in our discussion as a weakness that the study had an exclusively European genetic ancestry, but we have now clarified the reasons for this and called for genetic data collection and inclusion of studies in different genetic ancestries.

On page 7, we write: "[...] since twin data with DNA in other ancestries were not available in sufficient sample sizes. Our findings may not be generalisable to non-European genetic ancestries. The current study underlines the utility of twin data with DNA and should encourage funding for genetic data collection in multi-ancestry twin cohorts."

- 3- genetic correlations: would it be possible to show some genetic correlations between the results of this paper's GWAS summary statistics and other (mean or variance) GWAS? For example the mean GWAS for the same traits, or the population-based variance GWAS, etc. On page 5 the authors mentioned that they "could not estimate the genetic correlation between all phenotypes because the SNP-heritabilities were too low and imprecise" Is that true of genetic correlations with other well-powered GWAS as well? I would find those results much more informative than the two-sample MR.

A: There are currently no population-based variance GWAS for psychiatric phenotypes, but we did look at correlations with mean-based GWAS of the same traits for our three phenotypes, which had positive though statistically non-significant heritability estimates (e.g., ADHD .22, ASD .09 and Depression .03), while the relationships with their counterpart mean-GWAS was positive, only ADHD_MZ had significant genetic correlations (with ASD_PGC (.28) and Depression PGC (.30). See Panel B of figure below. However, we did not present them in the paper because of their imprecision. Interpreting the statistically significant correlations, we would conclude that genetic main effects on risk for depression are associated with genetics of greater

environmental sensitivity for ADHD traits. A similar pattern of associations exists for ASD genetic main effects.

- 4- mendelian randomization: I'm not sure the current setting satisfies the exclusion restriction assumption needed to perform MR. If I understand correctly, one would need to assume that certain genetic variants only influence directly the mean but only indirectly the variance of a phenotype, while actually testing that the mean of a phenotype has a causal effect on its variance. I would drop the MR analysis, or at least warrant the reader about the plausibility of the necessary assumptions in the current setting.

A: We used the summary data from the MZ GWAS as an outcome in Mendelian randomization analyses. Here, we investigate the hypothesis of whether differences in the mean of each exposure (e.g., depression) affect the variance in each psychiatric outcome. This analysis depends on the instrumental variable assumptions: 1) relevance, 2) independence and 3) exclusion. The exclusion restriction assumption requires that the SNPs only affect the variance in the trait via their effects on the mean levels of each exposure. This assumption is plausible, and we present sensitivity analyses to assess this assumption. This is a major innovation of our paper and illustrates how vQTL GWAS summary data from MZ twins can be used to investigate the causes of variance in a trait. We have described these assumptions in full and provided a full MR-STROBE checklist. We have now added this information to the manuscript.

On page 11 we write: “[...] We used the summary data from the MZ GWAS as an outcome in Mendelian randomization analyses. Here, we investigate the hypothesis of whether differences in the mean of each exposure (e.g., depression) affect the variance in each psychiatric outcome. This analysis depends on the instrumental variable assumptions: 1) relevance, 2) independence and 3) exclusion. The exclusion restriction assumption requires that the SNPs.

5- simulations: a few questions on the simulation

- given the paper's estimates of heritability for the phenotypes in question, it seems to me that figure 5 suggests that a population-based design with 20k individuals actually has more power. In other words, it might give more reliable results to use the MZ twins as if they were just random people instead of twins. Is that correct? if so, I think it should be pointed out somewhere in the text.

A: The x axis represents the heritability of "mean" levels of the trait, rather than the heritability of "variance". So, the power of MZ design to detect vQTL loci surpasses that of unrelated populations with a similar sample size (i.e. 20k individuals or 20k twins) when narrow-sense heritability of the trait is over .5. We have now modified the figure to prevent power being saturated which previously made comparisons difficult / potentially misleading. The y-axis shows the F-statistic of the vQTL association test (on the \log_{10} scale), and the x-axis shows the narrow sense heritability used in the simulation.

- instead of/decide that current graph in figure 5 I would suggest adding another one where heritability is fixed at the level estimated for the current phenotypes and what varies on the x-axis is the sample size of twins needed to achieve a certain statistical power. This would help future researchers who might want to replicate the current results once more twin data is available.

A: We appreciate the suggestion. However, we're not sure how to parameterise such power calculations without knowing likely vQTL effect sizes. The purpose of Figure 5 is to illustrate how vQTL estimation in MZ compares against population-based methods. We have now updated the y-axis scale in Figure 5a from statistical power to expected F-statistics (on the log scale) to prevent power from being saturated, which previously made comparisons difficult / potentially misleading.

- in the method section, I don't fully understand the notation of the equations on page 9. Shouldn't G_i also be indexed by the SNP, G_{ij} ? Shouldn't there be exclamation over different j SNP somewhere in the data generating model of the phenotype y ? Why is the mean of that $z_i = 0$, aren't there any average effects of the remaining polygenic risk? What is ϵ_i ? Is $\sigma^2_v = \beta^2 G_i$?

A: We have now simplified the annotation in the simulations to remove the j term, because we are only simulating one vQTL. The polygenic component is simply a normally distributed variable z_i which is identical within MZ sibship. For simplicity, we have fixed the mean of y to be 0 as this plays no part in the estimation. We have amended the text on pages 9 & 10.

Nature Human Behaviour

Institute of Psychiatry,
Psychology & Neuroscience
Social, Genetic & Developmental
Psychiatry Centre

14th March 2025

Ref: NATHUMBEHAV-24041546A

Dear [REDACTED]

We are delighted that our research article has been provisionally accepted for publication at Nature Human Behaviour, a journal we consider an ideal outlet for our findings. We have carefully considered your notes and incorporated the guidance from the author's check-list document into an updated manuscript. We have now modified the title to "**Genetics of monozygotic twins reveals the impact of environmental sensitivity on psychiatric and neurodevelopmental phenotypes**", as per your recommendation.

With regards to the final comments from reviewers, we would like to thank them for their feedback, and respond:

"Regarding the exclusion restriction – it's important here to be clear that the exclusion restriction requires that all of the effects of SNPs on the outcome must be mediated via the exposure – depression. It is possible that the SNPs affect variance in depression, independent of their effects on the mean levels of depression. However, given that these variants were discovered in standard mean GWAS, it is likely that they at least partly impact variance through mean effects. If the SNPs do have large vQTL effects on depression that are independent of their effects on mean liability to depression, then our MR estimates could be biased in either direction depending on the direction of effects vQTL on the outcomes.

Regarding why it is important to use MR to show causality: we ran a simulation to check for obvious violation of IV3 being auto-induced somehow. This obviously doesn't rule out more obscure mechanisms of IV3 violation, but the implications for MR results are:

- Type I error is well controlled for in this data generating model.
- There are MZ variance effects that are likely undetectable at GWAS significance thresholds but are revealed when hypothesising that there are variance effects at the main effects. Essentially it's a mechanism to improve power to detect evidence of vQTLs.

Overall, MR analyses suggest that higher liability to depression increase environmental sensitivity to depression. For a phenotype such as depression with hypothesised but inconsistently evidenced GxE effects, our study lends weight to the GxE theories of individual differences in depression. Furthermore, MR allows us to estimate the magnitude of effect of mean differences in liability to traits like depression on environmental sensitivity on an interpretable scale. Furthermore, MR allows us to estimate the magnitude of effect of mean differences in liability to traits like depression on environmental sensitivity on an interpretable scale."

We have now updated the manuscript in the discussion of results in page 12, with a note on how this violation of MR assumption could impact interpretation of results.

In the authors' checklist document, we were asked to also provide a brief (250 characters) summary of the research, and senior authors' and institutions' social media handles for linking to post following publication. These information are provided below:

"This study uncovered several genetic associations linked to environmental sensitivity in psychiatric and neurodevelopmental traits, in an international collaboration using data from over 21,000 monozygotic twins, making it the largest genetic study of its kind."

Institutional and authors' media handles are as below:

Prof Patricia Munroe:

@munroe_patsy

@munroepb.bsky.social

QMUL handles : @QMULWHRI, @QM_SBBS

Prof Thalia Eley:

@thaliaeley.bsky.social

@thaliaeley

Kings' handle: @SGDPCentreKCL

Prof Neil Davies:

@neilmdavies.bsky.social

@nm_davies

UCL's handles: @uclpsychiatry.bsky.social, @uclpsychiatry

We hope we have satisfactorily addressed all required edits.

Thank you again for accepting our research paper for publication.

Yours sincerely,

Dr Elham Assary, on behalf of the authors
Social, Genetic and Developmental Psychiatry Centre,
Institute of Psychiatry, Psychology and Neuroscience,
King's College London, London, UK
Email: elham.1.assary@kcl.ac.uk